

Comparing SWAT with SWAT-MODFLOW hydrological simulations when assessing
the impacts of groundwater abstractions for irrigation and drinking water
Wei Liu[1], Seonggyu Park[2, 3], Ryan T. Bailey[2] , Eugenio Molina-Navarro[1, 4], Hans Estrup Andersen[1],
Hans Thodsen[1], Anders Nielsen[1], Erik Jeppesen[1], Jacob Skødt Jensen[5], Jacob Birk Jensen[6, 7] and
Dennis Trolle[1].
[1]Department of Bioscience, Aarhus University, Silkeborg, Denmark;
[2]Department of Civil and Environmental Engineering, Colorado State University, Fort Collins,
Colorado, USA;
[3]Blackland Research & Extension Center, Texas A&M AgriLife, Temple, United States;
[4]Deparment of Geology, Geography and Environment, University of Alcalá. Alcalá de Henares,
Madrid, Spain.
[5]NIRAS, Aarhus, Denmark;
[6]Department of Civil Engineering, Aalborg University, Aalborg, Denmark;
[7]WatsonC, Aalborg, Denmark.
Correspondence: Wei Liu (weli@bios.au.dk, liuwei.alan@gmail.com)
Key Points:
• We compared the performance of SWAT and SWAT-MODFLOW and assessed the simulated

18        streamflow signals in response to a range of groundwater abstraction scenarios targeted for

19        irrigation and drinking water.

• The SWAT-MODFLOW complex was further developed to enable the application of the Drain

21        Package and an auto-irrigation routine.

• A PEST-based approach was developed to calibrate the coupled SWAT-MODFLOW.
• The SWAT-MODFLOW model produced more realistic results on groundwater abstraction

24        effects on streamflow.



**Abstract**

Being able to account for temporal patterns of streamflow, the distribution of groundwater resources, as well as the interactions between surface water and groundwater is imperative for informed water resources management. We hypothesize that, when assessing the impacts of water abstractions on streamflow patterns, the benefits of applying a coupled catchment model relative to a lumped semi-distributed catchment model outweigh the costs of additional data requirement and computational resources. We applied the widely used semi-distributed SWAT model and the recently developed SWAT-MODFLOW model, which allows full distribution of the groundwater domain, to a Danish, lowland, groundwater-dominated catchment, the Uggerby River Catchment. We compared the performance of the two models based on the observed streamflow and assessed the simulated streamflow signals of each model when running four groundwater abstraction scenarios with real wells and abstraction rates. The SWAT-MODFLOW model complex was further developed to enable the application of the Drain Package of MODFLOW and to allow auto-irrigation on agricultural fields and pastures. Both models were calibrated and validated, and an approach based on PEST was developed and utilized to enable simultaneous calibration of SWAT and MODFLOW parameters. Both models demonstrated generally good performance for the temporal pattern of streamflow, albeit SWAT-MODFLOW performed somewhat better. In addition, SWAT-MODFLOW generates spatially explicit groundwater-related outputs, such as spatial-temporal patterns of water table elevation. In the abstraction scenarios analysis, both models indicated that abstraction for drinking water caused some degree of streamflow depletion, while abstraction for auto-irrigation led to a slight total flow increase (but a decrease of soil or aquifer water storages, which may influence the hydrology outside the catchment). In general, the simulated signals of SWAT-MODFLOW appeared more plausible than those of SWAT, and the SWAT-MODFLOW decrease in streamflow was much closer to the actual volume abstracted. The impact of drinking water abstraction on streamflow depletion simulated by SWAT was unrealistically low, and the streamflow increase caused by irrigation abstraction was exaggerated compared with SWAT-MODFLOW. We conclude that the further developed SWAT-MODFLOW model calibrated by PEST had a better hydrological simulation performance, wider possibilities for groundwater analysis, and much more realistic signals relative to the semi-distributed SWAT model when assessing the impacts of groundwater abstractions for either irrigation or drinking water on streamflow; hence, it has the potential to be a useful tool in the management of water resources in groundwater-affected catchments. However, this comes at the expense of higher computational demand and more time consumption.



## 1. Introduction

The interaction between groundwater and surface water is an important aspect of the water cycle, and the management or use of one often impacts the availability and temporal patterns of the other. Improper management and over-exploitation of these water resource components influence the sustainability of both the water resource itself and also the ecosystems that it supports. Groundwater abstraction can cause a decline of the water table, and it thereby directly affects surface water bodies connected to the aquifer (Jeppesen et al., 2015; Vainu and Terasmaa, 2016; Stefania et al., 2018). For rivers in which a considerable portion of the streamflow is base flow, this can have a strong influence on the general flow and deteriorate the function of river ecosystems (Johansen et al., 2011; Pardo and Garcia, 2016). However, interactions between groundwater and surface water are difficult to observe and measure, and it is, therefore, difficult to determine how much of the reduced streamflow recorded in some rivers is due to abstractions and how much is due to natural weather-induced variability in water table elevation.

For quantitative assessment of the impacts of pumping wells on streamflow, a hierarchy of modeling tools has been developed, ranging from analytical models based on simple water balance equations to regional, three-dimensional numerical models, depending on the complexity and available data source of the site (Chen and Yin, 2001; Parkin et al., 2007; May and Mazlan, 2014). Analytical models generally require less data for parameter identification and may therefore be applied when available data are sparse, thus offering water managers a simple approach for estimating streamflow depletion with less time, expertise, and financial costs (Glover and Balmer, 1954; Hunt, 1999; Huang et al., 2018; Zipper et al., 2018). Nevertheless, as they do not simulate many of the physical processes and ignore the real-world complexity, they may render unrealistic results. In contrast, numerical, process-based models consider the entire complexity and heterogeneity of river-aquifer systems. Such models can simulate the regional groundwater dynamics as well as the interactions between groundwater and surface water. They are therefore part of local water management applications including estimation of streamflow depletion, although they are generally more time-consuming and costly to set up, calibrate, test, and apply.

MODFLOW is a physically-based, fully-distributed, and three-dimensional (3D) finite-difference groundwater model, and it is considered a state-of-the-art international standard for simulating and predicting groundwater conditions (http://water.usgs.gov/ogw/modflow/). It can be used to simulate both steady state and transient conditions. MODFLOW outputs include groundwater hydraulic head or drawdown at the center of each grid cell as well as groundwater flow rates to/from each stream



segment if the River (RIV) Package or the Streamflow Routing Package (SFR) is used (Wei et al.,
2018). A number of studies have applied MODFLOW to assess the impact of groundwater abstraction
on surface water resources (Sanz et al., 2011; May and Mazlan, 2014; Shafeeque et al., 2016; Stefania
et al., 2018). However, MODFLOW does not simulate surface processes such as land-atmosphere
interactions, agricultural management practices, and surface runoff (Lachaal et al., 2012; Surinaidu et
al., 2014). To obtain spatial-temporal varying recharge rates, MODFLOW is therefore often linked
with land-surface models such as the Precipitation-Runoff Modelling system (Markstrom et al., 2008;
Markstrom et al., 2015) and the Soil and Water Assessment Tool (SWAT) (Izady et al., 2015; Wei et
al., 2018).
The SWAT model is a semi-distributed catchment-scale model and has been widely used to simulate
surface runoff, sediment erosion, pesticide and microorganism transport, and nutrient cycling in
catchments at different geographical locations and scales (Nielsen et al., 2013; Fukunaga et al., 2015;
Malago et al., 2017; Liu et al., 2019). In SWAT, the basin is divided into subbasins through a
topography-based delineation, each subbasin containing a tributary of the river. Each subbasin is
further divided into Hydrologic Response Units (HRUs), which are unique combinations of land use,
soil type, and surface slope. When simulating hydrological dynamics, the areas of the HRUs are
lumped within each subbasin, which makes SWAT computationally very efficient, but this comes at
the expense of losing the spatial discretization of HRUs within a subbasin. SWAT has been utilized to
simulate and quantify the groundwater resources (Ali et al., 2012; Cheema et al., 2014; Shafeeque et
al., 2016) or the effects of drinking water or irrigation pumping on streamflow (Güngör and Göncü,
2013; Lee et al., 2006). However, the SWAT model has traditionally emphasized surface processes as
the model only includes a relatively simple representation of groundwater dynamics, and its output
does not give any spatially explicit information on the groundwater table. In the most recent version
of SWAT (v. 670), groundwater is represented by a lumped module in individual subbasins divided
into a shallow and a deep aquifer. Both the shallow and the deep aquifer may contribute to streamflow
as baseflow through a linear reservoir approximation, ignoring distributed parameters such as
hydraulic conductivity and storage coefficients (Kim et al., 2008). With this simplified implementation
of groundwater dynamics in SWAT, the model can mislead evaluation of groundwater resources or
perform rather poorly in catchments where the streamflow is strongly dependent on groundwater
discharge (Gassman et al., 2014).
To the best of our knowledge, there are two main approaches for making SWAT perform better in
groundwater-dominated catchments. One approach is to modify the SWAT groundwater module code



itself. For example, (Zhang et al., 2016) modified the subroutines in the SWAT source code by
converting the shallow aquifer water storage change into water table fluctuation with three
groundwater parameters added, namely specific yield, the bottom bed burial depth, and shallow aquifer
porosity. The modified SWAT could then simulate both water table fluctuations and water storage of
the shallow aquifer in time and space. However, it still applied a lumped, linear reservoir approach to
simulate groundwater storage and derive the water table at HRU level, which could give rise to errors
as the HRUs are not spatially explicit within a subbasin. (Pfannerstill et al., 2014) implemented a three-
storage concept in the groundwater module by splitting the shallow aquifer into a fast and a slow
contributing aquifer. (Nguyen and Dietrich, 2018) replaced the deep aquifer in the original SWAT
model with the multicell aquifer model. In both of these studies, the modified SWAT model achieved
a better prediction of baseflow than the original SWAT model. However, both models only improved
a part of aquifer system simulation, either the shallow aquifer or the deep aquifer. In addition, they
maintained the semi-distributed approach.
The other approach for improving the performance of SWAT in groundwater-dominated catchments
is to couple SWAT with a physically based, spatially distributed numerical groundwater model, such
as MODFLOW. There are a few studies that have tried to integrate SWAT and MODFLOW code into
one model complex (Kim et al., 2008; Yi and Sophocleous, 2011; Guzman et al., 2015; Bailey et al.,
2016). The most recent of these, the SWAT-MODFLOW code developed by (Bailey et al., 2016)
couples the most recent SWAT code with the MODFLOW-NWT code (a Newton-Raphson
formulation for MODFLOW-2005 (Niswonger et al., 2011), which improves the solution of
unconfined groundwater-flow problems). This coupled version has several advantages over others: an
efficient HRU-grid cell mapping scheme (including generation of geographically explicit HRUs), the
ability to use SWAT and MODFLOW models of different spatial coverage, the use of the
MODFLOW-NWT code, public availability, and a graphical user interface that has been recently
developed for its application (Bailey et al., 2017; Park et al., 2018). Recently, the current published
SWAT-MODFLOW code (Version 2 on the SWAT website) has been applied to catchments in the
USA (Bailey et al., 2016; Abbas et al., 2018; Gao et al., 2019), Canada (Chunn et al., 2019), Denmark
(Molina-Navarro et al., 2019), Iran (Semiromi and Koch, 2019), and Japan (Sith et al., 2019). It has
also been further developed for application in large-scale mixed agro-urban river basins (Aliyari et al.,
2019). Within the coupled SWAT-MODFLOW framework, SWAT simulates surface hydrological
processes, whereas MODFLOW-NWT simulates groundwater flow processes and all associated
sources and sinks on a daily time step. In addition, the HRU-calculated deep percolation from SWAT
is passed to the grid cells of MODFLOW as recharge, and MODFLOW-calculated groundwater-



surface water interaction fluxes are passed to the stream channels of SWAT. Hence, the model complex
accounts for two-way interactions between groundwater and surface waters and thereby enables a
potentially much better representation and thus understanding of the spatial-temporal patterns of
groundwater-surface water interactions, which are of key importance to catchment management in
groundwater-dominated catchments.
In Denmark, approximately 800 million $m^3$ of water are abstracted annually and used for irrigation or
drinking water (GEUS, 2009), making the country highly dependent on groundwater. Since the very
dry summers in 1975 and 1976 led to dry out of many watercourses around some cities in Denmark,
the national government has endeavored to regulate the abstraction of surface and groundwater to a
level preventing negative impacts on in-stream biota. Gradually, direct abstraction from surface waters
has been prohibited and groundwater abstraction is regulated to secure a certain minimum flow in all
Danish rivers, mainly by moving the abstraction wells away from riverbanks and wetlands and
implementing a groundwater abstraction permit authority system. However, there still remains some
areas where groundwater exploitation is above the sustainable yield and causes streamflow depletion
according to the national water resource model (Henriksen et al., 2008).
To better understand how abstraction wells used for drinking water or irrigation may influence nearby
streamflow, we applied both SWAT and SWAT-MODFLOW to a lowland catchment in Northern
Denmark – the Uggerby River Catchment. We hypothesize that, when assessing impacts of water
abstractions on streamflow patterns, the benefits of applying SWAT-MODFLOW relative to SWAT
outweigh the costs of additional data requirements and computational resources. We compared the
performance of the two models and assessed the simulated signals of streamflow in a range of
groundwater abstraction scenarios with real wells and abstraction rates for either drinking water or
irrigation with both models. The SWAT-MODFLOW complex used in this study was further
developed based on the publically available version (https://swat.tamu.edu/software/swat-modflow/)
to enable application of the Drain Package of MODFLOW and to allow auto-irrigation. In addition, an
approach based on PEST (Doherty, 2018) was developed to calibrate the coupled SWAT-MODFLOW
by adjusting SWAT and MODFLOW parameters simultaneously against the observations of both
streamflow and groundwater table.




## 2. Materials and methods

### 2.1 Study area

The Uggerby River Catchment lies between latitude 57°17′10"- 57°35′25" N and longitude 9°58′47"- 10°19′55" E. It covers an area of 357 km$^2$ and is located in the Municipality of Hjørring, which is situated in the northern part of Jutland, Denmark (Fig. 1). The Uggerby River originates from the southern part of Hjørring in Sterup and winds its way through the area of Hjørring, Sindal, Mosbjerg, Bindslev, and Uggerby and then discharges into the bay Tannisbugten at the coast of the North Sea. The study area has a typical Atlantic climate, which is temperate with an average annual temperature around 8 °C, being warmest in August (17 °C average) and coldest in January (0.5 °C average). The average annual precipitation during the study period 2002-2015 was approximately 933 mm with no obvious distinctions among seasons.

**Figure 1.**

The mean catchment elevation is 34.5 m a.s.l and ranges from 0 to 108 m. Land cover in the catchment is dominated by agricultural land, and the other land use types include evergreen forest, pasture, wetland, and urban areas. The soil types are loamy sand, sandy loam, and sand. The main crops grown in the area include winter wheat, winter rape, barley, corn, and grass. Artificial tile drains have been installed in parts of the agricultural land in the catchment, although the precise drainage locations are somewhat uncertain (Olesen, 2009). According to an investigation carried out by Hjørring Municipality in 2009, there are 101 drinking water pumping wells registered within the catchment and 57 irrigation pumping wells placed on pasture and agricultural land. Generally, irrigation only occurs from April to October. The average annual irrigation amount varies from 80 to 200 mm depending on the types of crop and soil conditions (Aslyng, 1983).

### 2.2 Model set-up and coupling

#### 2.2.1 SWAT model set-up

We used the QSWAT 1.5 interface (George, 2017), which works with the latest SWAT Editor version 2012.12.19 and is integrated into a QGIS 2.8.1 interface. The input data for the SWAT model in this study include topography, land use, soil, climate, agricultural management, wells, and wastewater discharge as point sources.



The catchment was divided into 19 subbasins (Fig. 1) based on the 32 m pixel size Digital Elevation
Model (DEM), which has been resampled from a 1.6 m LIDAR DEM (Knudsen and Olsen, 2008). For
the creation of HRUs, we used the land use map based on the Danish Area Information System (NERI,
2000) and the soil map based on a national three-layer soil map with a 250 m grid resolution (Greve
et al., 2007), and surface slope type was classified into three classes (<2%, 2-6%, >6%). To reduce the
number of HRUs and facilitate the posterior model linkage process, land use for range-grasses and
range-brush, which covered only 1.3% and 1.9% of the total catchment area, respectively, were merged
into pasture, and water (0.9%) was merged into wetland areas. In order to represent the agricultural
management practices in detail, the agricultural area was split into three equally sized types with
different five-year crop rotation schedules (Table 1) based on the real contour of agricultural field plots
and the land use map. Similar to land use, soil types covering a minor part of the catchment (1% or
less) were merged into similar soil types. The distribution and proportion of each land use, soil type,
and slope band after reclassification are shown in Fig. 2. Based on the combination of land use, soils,
and slope, the catchment was discretized into 2620 HRUs.
**Figure 2.**
Climate data used in the model comprised the 10 km-grid national daily precipitation data (six stations
inside the catchment), 20 km-grid daily solar radiation and wind speed data (five stations inside or near
the catchment), gauged-level daily maximum and minimum temperatures, and relative humidity data
(one station, 27 km from the catchment ) during 1997-2015 from the Danish Meteorological Institute
(Lu et al., 2016).
Farm type and manure/mineral fertilizer application of each agricultural rotation as well as dates of
sowing, harvesting, and tillage were assigned based on reported statistics for 2005 available from
http://www.dst.dk/en (Table 1). We do not know the specific tile drain distribution within the entire
catchment. In general, loamy soils in relatively flat areas are known to be tile drained in Denmark
(Olesen, 2009). To represent this situation, tile drains were set up in agricultural land with a slope less
than 2% and for soil types with a clay content above 8% (Thodsen et al., 2015), representing 27% of
the agricultural land in the catchment.
We assumed that irrigation only occurs in the HRUs where irrigation pumping wells exist (based on a
MODFLOW model created by NIRAS A/S). It is difficult to know the exact dates and water amount
used for irrigation. Thus, to simulate the irrigation, auto-irrigation management was set up based on
heat unit scheduling for the HRUs containing irrigation pumping wells. For the auto-irrigation of crops,





the water resource used for pumping was defined as the shallow aquifer, and the soil water content,
commonly used as an indicator in actual field irrigation (Chen et al., 2017), was selected as the water
stress identifier with 70 mm as the initial water stress threshold. With the number and location of
pumping wells as well as their pumping rates obtained from the Well Package in the MODFLOW
model, the water abstraction amounts from drinking water wells were added up in each subbasin and
set as the water use pumped from the shallow aquifer in SWAT.
The only significant point source of the study area is the discharge from the wastewater treatment plant
in Sindal located in subbasin 16. With a few other minor sources aggregated to a total discharge from
the wastewater treatment plant, a total of 2768.8 $m^3$ of water was discharged into the stream per day
(data is based on an average for the period 2007-2010).

### 254    2.2.2 MODFLOW-NWT model set-up

A steady-state version of the MODFLOW-NWT model has previously been set up for the entire
Hjørring Municipality, covering an area of 930 $km^2$, in which the Uggerby River Catchment is situated
(Fig. 3). The model set-up was firstly established in 2011 and then updated in 2016 by the consultant
company NIRAS A/S and Hjørring Water Supply Company, and has been applied for water resources
management in the Hjørring Municipality. In the model set-up, the geology is represented by 5 hydro-
stratigraphic layers, discretized into 183,112 grids (376 rows and 487 columns) with a discretization
of 100 x 100 meters. The uppermost layer is unconfined and the remaining four layers are confined.
The Upstream Weighting (UPW) Package for MODFLOW, which contains hydraulic properties of
each cell, was used as the internal flow package, and a number of boundary condition packages,
including Time-variant specified-head Package, Drain Package, River Package, Well Package, and
Recharge Package, were employed in the model to simulate external stresses. The steady-state model
was calibrated using 1,063 head observations sampled during the period 1996-2010 at 1,006 well
locations distributed within the first, third, and fifth layer by a combination of manual calibration and
auto-calibration through PEST (http://www.pesthomepage.org/).
**Figure 3.**
Eighteen different hydraulic conductivity values exist in the originally calibrated MODFLOW model.
In order to facilitate the posterior SWAT-MODFLOW calibration, we reclassified and grouped the
specific hydraulic conductivities into five groups. The grouping was made for grid cells of similar
specific hydraulic conductivities, representing the sedimentary materials of clay, silt, silty sand,





mixture of silty sand and clean sand, and clean sand, respectively. Each group was assigned a unique
specific hydraulic conductivity, which could be targeted for calibration.
For the SWAT-MODFLOW set-up, we converted the modified calibrated steady-state model into a
transient model by assigning values to the specific yield (only for the unconfined layer) and specific
storage of each cell according to the type of sedimentary materials of the cell and representative values
of storage coefficients. The simulated heads generated by the steady-state model were used as the
initial head conditions for the transient model.
**2.2.3 SWAT-MODFLOW coupling**
SWAT and MODFLOW were combined using the coupling framework developed by (Bailey et al.,
2016) and following the procedures described in the instructions available from the SWAT website
(http://swat.tamu.edu/software/swat-modflow/).
For this study, the following changes were made to the original SWAT-MODFLOW code: (1) the grid
cells in the Drain Package were linked with SWAT subbasins so that groundwater removed from
subsurface drains is routed to stream channels; and (2) groundwater pumping in agricultural areas or
pastures is dictated by irrigation applied to HRUs through SWAT's auto-irrigation routines. For the
latter, this is achieved by calculating the daily volume of applied irrigation water (irrigation depth *
HRU area) and then extracting this volume from the underlying grid cells using MODFLOW's Well
Package (Fig. 4). When applying the Drain Package of MODFLOW, the original tile drain routine in
SWAT was disabled. The steps in the coupling procedure included: 1) disaggregation of HRUs to
disaggregated hydrologic response units (DHRUs) through GIS processing to make the model spatially
explicit; and 2) creation of six linking text files (HRUs to DHRUs, DHRUs to MODFLOW grids,
MODFLOW grids to DHRUs, MODFLOW river cells to SWAT subbasin rivers, MODFLOW drain
cells to subbasin rivers, irrigation pumping wells in HRUs to MODFLOW grids) through GIS
processing. All related files (MODFLOW input files, original SWAT model files, linkage files) were
stored in one working directory for SWAT-MODFLOW execution.
**Figure 4.**
**2.3 Model calibration**
**2.3.1 SWAT calibration**





The Sequential Uncertainty Fitting Algorithm (SUFI2), which is implemented in the SWAT-CUP
software (Abbaspour, 2015), was used to calibrate discharge performance in SWAT. The latest
SWAT-CUP version (5.1.6.2) was used. Calibration was performed based on daily discharge records
from 1 Jan. 2002 to 31 Dec. 2008, with a previous 5-year model warm-up period and using Nash-
Sutcliffe efficiency ($N_{SE}$) as the objective function. Five parameters at basin-wide level and 17
parameters at subbasin level related to streamflow were selected and assigned initial calibration value
ranges based on expert judgement and previous SWAT applications in Danish catchments (Lu et al.,
2015; Molina-Navarro et al., 2017).
In the study area, two hydrologically connected monitoring stations are found, located at the outlet of
subbasin 13 (station A) and subbasin 18 (station B), respectively (Fig. 1). The two stations represent a
small (average discharge 1.95 $m^3 s^{-1}$) and relatively large (average discharge 4.56 $m^3 s^{-1}$) stream in
Denmark, and both were used for calibration and validation in this study. Station A is located upstream
from station B and its flow therefore has an influence on station B. Thus, the simulated discharge of
station A was preliminarily calibrated first (initial range of related parameters are shown in Table 2),
running 3 iterations with 500 simulations each. After the final iteration for station A, the subbasin level
parameters for the area upstream station A were fixed, while the final ranges of the basin-wide
parameters were used in the subsequent calibration of station B. As the basin-wide parameter values
can impact the hydrology of the entire catchment, for the calibration of station B, discharge data from
both station A and B were included in the objective function. An additional three iterations with 500
simulations were run, where the subbasin level parameters for the remaining area upstream station B
were calibrated using the same initial parameter range as for station A (Table 2), while the basin-wide
parameter ranges from the final calibration step for station A were used as initial ranges. By this
approach, we attempted to make the basin-level parameters representative for both upstream and
downstream areas. Afterwards, the water stress threshold was calibrated manually to ensure proper
simulation of the annual irrigation amount, which ranges from 80 to 120 mm $yr^{-1}$ and occurs in the
period April to October (Aslyng, 1983). Once the calibration was completed and the parameters were
fixed, we validated the model by running one simulation from 1 Jan. 1997 to 31 Dec. 2015 using the
first 12-years as a warm-up period.
To analyze parameter sensitivity and make the sensitivity analysis comparable with SWAT-
MODFLOW, an additional iteration with 500 simulations was run for the calibration period. In this
iteration, the ranges of basin level parameters and subbasin level parameters for the area upstream
station A were the same as those in the final calibration step for station A, while the ranges of subbasin



level parameters for the area upstream station B were identical with the final calibration step for station
B.
**Table 2.**
**2.3.2 SWAT-MODFLOW calibration**
After model coupling, the SWAT-MODFLOW was calibrated by adjusting SWAT and MODFLOW
parameters simultaneously against the observations of both streamflow and groundwater table through
a combination of manual calibration and auto-calibration by the widely used PEST approach (Doherty,
2018). The periods used for model warm-up, calibration, and validation were identical to those used
for SWAT. SWAT-MODFLOW can also be run through SWAT-CUP, whereby the summary statistics
of model performance can be derived and directly compared between SWAT and SWAT-MODFLOW.
In addition, model.in and Swat_Edit.exe, which are included in the creation of the SWAT-CUP project
folder, can be used to adjust SWAT parameters within the PEST routine.
The framework using PEST to calibrate SWAT-MODFLOW was firstly introduced by (Park, 2018).
We applied this framework to this study as well but with BEOPEST (Doherty, 2018) instead of PEST
as the PEST executable file. Figure 5 presents a schematic diagram illustrating how PEST is utilized
for the SWAT-MODFLOW calibration in this study. Five types of files are required to run PEST:
PEST control file, PEST executable file, model batch file, model input template files, and model output
instruction files. The PEST control file is a master file that contains control variables, initial values
and ranges of model parameters, observations and their weights for deriving the value of the objective
function, as well as names of all input and output files related to calibration. BEOPEST was used as
the PEST executable file that enables parallelization of model runs on multiple computer cores, thereby
shortening the calibration time considerably. After each iteration of a PEST run, the PEST optimization
algorithm adjusts the model parameter values to optimize the value of the objective function. The
newly updated model parameter values are then written to model input files using input template files
and Swat_Edit.exe. Next, the SWAT-MODFLOW executable is called by a batch file and generates a
set of output files if the model runs successfully. A python script (exsimvalue.py) extracts the
simulated values from the streamflow output file (output.rch) and the groundwater table output file
(swatmf_out_MF_obs). The extracted simulated data are read by PEST using information from the
model output instruction file and then compared against the corresponding observations. Each iteration
includes a number of model runs according to the control variable set in the PEST control file to allow
adjustment of parameter values. After each iteration, the objective function and a Jacobian matrix are


calculated, based on which the PEST will make its decision for the next iteration until one of its
stopping criteria, specified in the PEST control file, is met. More detailed information about the
optimization process and principles of PEST can be found in (Zhulu, 2010) and the PEST manual
(Doherty, 2018).
**Figure 5.**
As shown in Table 3, 26 parameters from SWAT related to surface hydrological processes and 13
parameters from MODFLOW were selected and calibrated through PEST. For SWAT parameters,
with the parameters related to tile drains and groundwater excluded, the final calibrated parameter
values used in SWAT were applied as the initial values in PEST, and the parameter ranges used in the
iteration for SWAT parameter sensitivity analysis were employed as the parameter ranges in PEST.
By manually adjusting MODFLOW parameter values to test their impacts on model outputs, storage
coefficients (SY and SS), horizontal hydraulic conductivity (HK), and two drain conductance (COND)
were deemed as the potential sensitive parameters, with the value of HANI (the ratio of hydraulic
conductivity along columns to hydraulic conductivity along rows) always being 1 and the values of
VKA (the ratio of horizontal to vertical hydraulic conductivity) fixed as the values in the original
MODFLOW set-up. For MODFLOW parameters, the originally calibrated and modified parameter
values in the steady-state MODFLOW version were used as the initial parameter values in PEST, and
a small range around the initial values was assigned as the parameter range according to the experience
from manual calibration and representative values (derived from http://www.aqtesolv.com/aquifer-
tests/aquifer_properties.htm).
**Table 3.**
The observed streamflow used for calibrating SWAT-MODFLOW was identical to that used for
calibrating SWAT. Relatively continuous observations of the groundwater table were available at the
location of two grid cells, and these were used for calibrating the variation of the groundwater table
simulated by SWAT-MODFLOW. Because station A is located upstream from station B and its flow
thus has an influence on station B, the weight for deriving the objective function for station A, which
represents a small stream, was set to 2, and the weight for station B was set to 1. The weights for the
two grid cells were set to 1.
In order to establish template files and facilitate the process of modifying parameter values (HK, SS,
SY) in the UPW Package while running PEST, the parameter value file (PVAL) and Zone file





(https://water.usgs.gov/ogw/modflow-nwt/MODFLOW-NWT-Guide/) were first established based on
the original UPW Package through running a code file in FORTRAN.
Ten iterations were specified as the stop criteria in the PEST control file. Due to the large number of
grid cells (183,112) in the MODFLOW set-up and the amount of disaggregated HRUs (DHRUs,
66,765) compared with the case study conducted by (Bailey et al., 2017), it takes the coupled SWAT-
MODFLOW model complex around 4 hours to run a single simulation (12 years' simulation) when
MODFLOW runs with a daily interval. To shorten the calibration time, 11 BEOPEST slaves were
created on three computers with BEOPEST as the pest executable file so that 11 simulations could be
run simultaneously. A total of 638 simulations were run before the stop criteria was achieved. With
the calibrated parameters fixed, the water stress threshold was calibrated manually to ensure proper
simulation of the annual irrigation amount (ranging from 80 to 120 mm yr$^{-1}$, occurring in the period
between April to October) and make the simulated average annual irrigation amount in the irrigated
HRUs (mm yr$^{-1}$) comparative with that in the calibrated SWAT model. Finally, the SWAT-
MODFLOW model performance was validated following a procedure equivalent to that used for
SWAT.
**2.4 Water abstraction scenarios**
Besides the 158 wells registered within the Uggerby River Catchment, another 256 wells exist outside
the catchment but inside Hjørring Municipality (Fig. 3). All these wells were included in the Well
Package in the SWAT-MODFLOW set-up. In SWAT-MODFLOW, the irrigation pumping source was
defined as the third layer. For drinking water wells, 7 of the 101 drinking water wells were placed in
the first layer, 91 in the third layer and 3 in the fifth layer. In order to evaluate the impacts of both
irrigation and drinking water abstractions on streamflow for streams of difference sizes, four
abstraction scenarios were designed and applied to the Uggerby River Catchment using both models:
1) the no-wells scenario, where all abstractions are terminated; 2) the irrigation-wells-stop scenario,
where all abstractions in irrigation wells are terminated, while abstractions in drinking water wells
remain; 3) the drinking-wells-stop scenario, where all abstractions in drinking water wells are
terminated, while abstractions in irrigation wells remain; and 4) the baseline scenario, where
abstractions in all wells are included, which represents the current level of abstraction. We assumed
that the point source discharge to the stream in subbasin 16 would remain the same in all scenarios.
Once the scenarios were simulated, their impacts on streamflow were analyzed by assessing the
average annual runoff amount, the contribution of water balance components, and the temporal



dynamics of streamflow. The simulated signals of SWAT and SWAT-MODFLOW in the abstraction
scenarios were then compared.

**3 Results**

**3.1 Steady-state MODFLOW performance**

Visualization of the proximity of simulated and observed head contours (Fig. 6) was used to evaluate
how well the modified calibrated MODFLOW model performed at steady state and three summary
statistics were used as indicators for goodness of model fit (Table 4). The simulated heads and
summary statistics have changed little compared with the original calibrated MODFLOW set-up. Thus,
the modified calibrated MODFLOW model was satisfactory and suitable as a basis for coupling to
SWAT in transient mode.

**Figure 6.**

**Table 4.**

**3.2 SWAT and SWAT-MODFLOW transient model performance**

The SWAT and SWAT-MODFLOW models both represented well the streamflow hydrographs during
the calibration period, while during the validation period, one high peak flow event occurred in the
SWAT and SWAT-MODFLOW simulations but not in the observations (Fig. 7). The baseflow was
generally reproduced well by both models, but the SWAT-MODFLOW visibly performed better.

**Figure 7.**

**Table 5.**

Compared with the recommended evaluation criteria by (Moriasi et al., 2015), the statistical
performance (Table 5) suggested "very good" performance of both models during the calibration
period based on percent bias ($P_{BIAS}$). During the validation period, the models performed "good" at
station A and "satisfactory" at station B. For $N_{SE}$ values, the performance was "very good" for SWAT-
MODFLOW calibration at station B, "good" for SWAT-MODFLOW calibration at station A,
"satisfactory" for SWAT calibration and SWAT-MODFLOW validation at both stations and SWAT
validation at station A, but "unsatisfactory" for SWAT validation at station B. For $R^2$ values, the





performance was "good" for SWAT-MODFLOW calibration, "satisfactory" for SWAT calibration
and SWAT-MODFLOW validation, but "unsatisfactory" for SWAT validation.
The statistical performances of SWAT-MODFLOW with and without PEST calibration were
compared. After calibration by PEST, the summary statistics of SWAT-MODFLOW were improved,
especially for the validation period at station B where the performance increased from "unsatisfactory"
to "satisfactory" according to $N_{SE}$ values (Table 5). In addition, the weighted residuals between
simulation and observation were reduced after calibration by PEST, with the reduced residuals mainly
coming from streamflow simulation (Table 6).
**Table 6.**
In SWAT, almost all the top 12 sensitive parameters (Fig. 8) were surface process parameters (Table
2) except for the groundwater parameter GW_DELAY. In contrast, for SWAT-MODFLOW (Table 3),
all the top 12 sensitive parameters were groundwater parameters with the exclusion of only one surface
process parameter OV_N.
**Figure 8.**
Compared with SWAT, the SWAT-MODFLOW model not only produced output for streamflow but
also for the groundwater table of each cell on any given day. The variation of groundwater heads across
the simulation period was minimal for layer 1, while there was some, albeit small, variation in layer 3
(Fig. 9). There was generally a good agreement between the groundwater head level and dynamics
simulated by SWAT-MODFLOW and that was recorded at the two observation wells within the
catchment (Fig. 10).
**Figure 9.**
**Figure 10.**
For the water balance, the evaporation simulated by SWAT-MODFLOW was a little higher (13 mm
$yr^{-1}$) than that simulated by SWAT, while the total water yields (total stream flow) simulated by SWAT
and SWAT-MODFLOW were almost equal (Table 7). The water balance components, however,
differed substantially. Compared with SWAT, the surface runoff simulated by SWAT-MODFLOW
was a little higher, while the lateral subsurface flow and groundwater flow (simulated by the River
Package) were much lower. In SWAT-MODFLOW, the largest contributor to streamflow was the





drain flow simulated by the Drain Package (constituting 70% of the streamflow). Conceptually,
however, this can also be viewed as a surface-near groundwater contribution. Hence, when lumping
the contribution from drains and groundwater, these are clearly the dominant sources for streamflow
in both the SWAT and SWAT-MODFLOW model.
**Table 7.**
**3.3 Water abstraction scenarios simulation**
The annual abstractions by drinking water wells or irrigation wells set up in the two models were
approximately equivalent (Table 8). In the SWAT simulations, compared with the no-wells scenario
(scenario 1), a decrease in the average annual stream flow was observed in scenario 2 (only drinking
water wells), while an increase was recorded in scenario 3 (only irrigation wells) and scenario 4 (both
drinking water and irrigation wells). In the SWAT-MODFLOW simulations, the average annual
streamflow decreased not only in scenario 2, but also  in scenario 4, and at subbasin 18 outlet in
scenario 3, while a slight increase occurred at subbasin 13 outlet in scenario 3. The decrease in scenario
2 simulated by SWAT-MODFLOW was much larger than that by SWAT and also closer to the
abstracted amount, and the increase at subbasin 13 outlet in scenario 3 simulated by SWAT-
MODFLOW was apparently lower than that simulated by SWAT (Table 8).
**Table 8.**
In SWAT, the decrease of average annual total flow in scenario 2 was minimal as a result of a tiny
decrease in the groundwater return flow (Fig. 11a). In scenario 3 and scenario 4, with unchanged tile
flow, all the other flow components rose, especially groundwater and lateral soil discharge. In SWAT-
MODFLOW, the decrease of average annual total flow in scenario 2 also resulted from a decreased
groundwater return flow, but the decrease was much larger than that simulated by SWAT. In scenario
3, the lateral soil runoff and drain flow increased in SWAT-MODFLOW similar to SWAT, while in
scenario 4, reduced drain flow was recorded (Fig. 11b). Compared with the no-wells scenario, the
amount of evapotransportation remained unchanged in scenario 2, whereas it increased by 5 mm yr$^{-1}$
in the sceanarios with irrigation wells in both the SWAT and SWAT-MODFLOW simulations. In the
scenario with only irrigation, evaportransportation and total flow increased in both the SWAT and
SWAT-MODFLOW simulations, but the soil or aquifer water storage decreased according to the water
balance.



**Figure 11.**

When comparing the temporal patterns of streamflow with the no-wells scenario (scenario 1), we found the daily discharge difference in scenario 2 (only drinking water wells) to be almost always negative (sometimes zero), while in scenario 3 (only irrigation wells) and scenario 4 (both drinking water and irrigation wells) it fluctuated around zero in simulations by both SWAT and SWAT-MODFLOW (Fig. 12). Thus, the daily flow in the scenario with drinking water wells was almost always lower than the scenario without drinking water wells, and the daily flow in the scenario with only irrigation wells or the scenario with both irrigation and drinking water wells could be higher or lower than the scenario without wells. The daily discharge difference between scenario 2 and the no-wells scenario simulated by SWAT-MODFLOW was obvious, but the difference using SWAT was minimal. In the comparison of scenario 3 with the no-wells scenario, when the discharge difference was positive after an irrigation event, it descended smoothly in the SWAT simulation and more sharply in the SWAT-MODFLOW simulations.

**Figure 12.**

In the SWAT-MODFLOW set-up, the water exchange between aquifer and streams occurs between each MODFLOW river/drain cell and its surrounding cells. The newly developed SWAT-MODFLOW model complex can output the daily rate of water exchange between aquifer and streams for each subbasin. When the water exchange is positive, it is indicative of water flow from the aquifer to the stream. The temporal pattern of groundwater discharge was the same as for the stream flow, and the temporal patterns of the differences in groundwater discharge between the abstraction scenarios and the no-wells scenario were similar to the differences in streamflow, except for some peak flow days (Fig. 13), which indicates that the abstraction-induced streamflow change followed the groundwater discharge change.

**Figure 13.**

**4. Discussion**

**4.1 Performance and parameter sensitivity of SWAT and SWAT-MODFLOW**

Both the SWAT and SWAT-MODFLOW model simulated the temporal patterns of streamflow generally well at the two hydrological stations during the calibration and validation periods. However, visually SWAT-MODFLOW performed better, especially during recession curves and low flow


periods, suggesting a better simulation of the interaction between surface water and groundwater.
Accordingly, the corresponding summary performance statistics were also better for SWAT-
MODFLOW. The simulated peak flow on 16 October 2014 by both models was much higher than the
observed data (Fig. 7). This discrepancy may be attributed to a high record of precipitation on that day
based on a 10 by 10 km grid, which may not be representative for the wider catchment. Additionally,
it is also likely that the observed streamflow was underestimated as it is calculated from the Q-h
relation, which typically does not adequately cover peak flow events (Poulsen, 2013).
In the parameter sensitivity analysis, the surface process parameters of the two models shared the same
ranges, while the models had different groundwater modules and parameters. While the SWAT-
MODFLOW calibration was based on an objective function that took into account not only streamflow
but also groundwater heads at the location of two wells, the calibration by PEST mainly improved the
streamflow simulation performance (Table 4). According to the parameter sensitivity ranking, the
parameters regarding groundwater processes in SWAT-MODFLOW played an important role in the
streamflow simulation performance, while in SWAT, the impact of groundwater module parameters
on streamflow simulation was generally insignificant. This reflects the shortcoming of the SWAT
groundwater module, which ignores the variability in distributed parameters such as hydraulic
conductivity and storage coefficients, represents groundwater by a lumped module in individual
subbasins, and contributes to the stream network as baseflow based on a linear reservoir approximation.
With this simplified implementation of groundwater dynamics and water exchange between surface
water and groundwater in SWAT, the discharge simulated by SWAT cannot be optimized to the same
extent as that simulated by SWAT-MODFLOW.
The availability of spatial-temporal patterns of the groundwater head in SWAT-MODFLOW could
significantly benefit groundwater resources management and provide yet another level of
understanding of water resources dynamics within a catchment. The outputs of SWAT-MODFLOW
in this study showed that the model performed well, not only in streamflow simulations but also with
respect to the spatial-temporal patterns of the simulated groundwater head. In contrast, since no
information of groundwater table output is provided by SWAT, its goodness in streamflow simulation
may potentially be based on an improper groundwater simulation where its performance on
groundwater simulation is unknown.
**4.2 Models ability to simulate effects of groundwater abstractions on streamflow**





In scenario 2 where only drinking water wells are active according to the water balance where there is
no change in evaporation compared with the no-wells scenario, we expected that the streamflow
depletion simulated by SWAT would be approximately equivalent to the abstracted water volume,
taking into account a possible small change in the aquifer or soil storage. However, results in this study
showed that the impact of drinking water abstractions on streamflow in the SWAT simulation was
negligible. In the SWAT-MODFLOW set-up, because the aquifer in the Uggerby River Catchment is
connected to and interactive with an area outside of the topographical catchment (Fig. 3), the
abstraction from an aquifer located in the Uggerby River Catchment not only impacts the hydrology
inside but potentially also outside the catchment. According to the water balance, we expected that the
SWAT-MODFLOW simulated streamflow depletion in the catchment would be lower at a level
somewhat equivalent to the abstracted water volume. With equivalent abstraction for drinking water,
the annual flow decrease simulated by SWAT-MODFLOW was much larger than that by SWAT and
closer to the abstracted volume. Therefore, we conclude that SWAT simulations underestimate the
impacts of groundwater abstraction for drinking water on streamflow depletion, while SWAT-
MODFLOW provided more realistic assessments.
The simulated irrigation operation abstracts water from an aquifer and then applies the water onto the
surface of agricultural land or pasture. Most of the water infiltrates back into the soil and is then utilized
by the vegetation and partly lost through evapotranspiration or infiltrates deeper to the aquifer, and a
small part of the water might flow to streams directly as a small increase in surface runoff. Though the
abstraction causes groundwater depletion, the recharge from the irrigated water can partly refill the
aquifer and produce groundwater discharge. Since in the SWAT-MODFLOW set-up the aquifer in the
Uggerby River Catchment was connected and interactive with an outside area, after each event of
groundwater abstraction for irrigation, the aquifer storage would be recharged not only from the
irrigated land area but also by the groundwater flowing from the outside area. If the recharge rate is
larger than the abstracted water amount, the groundwater discharge to the stream will presumably
increase. Hence, the irrigation events also brought about a slight increase of average annual stream
flow at the subbasin 13 outlet (Table 8), and a slight total flow increase within the catchment (Fig.
11b). The subbasin aquifers in the SWAT set-up are closed and have no interaction with areas outside
a subbasin. Meanwhile, the abstracted amount of water from aquifers for irrigation is larger than the
amount of returning aquifer recharge from irrigated water, and we would therefore expect decreasing
groundwater discharge to streamflow in SWAT simulations. However, the SWAT simulations also
showed that irrigation led to enhanced streamflow (Table 8, Fig. 11a), which apparently was even
higher than the increase simulated by SWAT-MODFLOW. This supports the point mentioned above


that SWAT underestimates the abstraction effect on streamflow depletion. SWAT simulations can,
therefore, lead to incorrect assessments of the impacts of groundwater abstractions for irrigation on
streamflow, while SWAT-MODFLOW provided more realistic assessments.
Upon inspecting the SWAT source code, it appears that the groundwater discharge calculation
equation used in SWAT does not take into account the impact of water abstraction from shallow
aquifers on water table fluctuations. Thus, the groundwater removal by abstractions in the SWAT
simulation does not have a direct effect on the groundwater discharge, which may explain the
somewhat surprising simulation signals of SWAT. In addition, in the equation, the groundwater
discharge on the current day is highly related to the groundwater discharge on the previous day, and
the increase of the groundwater discharge resulting from each irrigation application could then lead to
enhanced groundwater discharge for several days in a row. This may explain why the increased
discharge following an irrigation event descended more smoothly in SWAT than in SWAT-
MODFLOW (Fig. 12).
In the SWAT-MODFLOW model, the exchange rate between groundwater and surface water is based
on the head difference between the river stage (or drain cell stage) and the head of its surrounding
groundwater grid cells. This can reflect the temporally dynamic hydrological processes and also the
impacts from all the external stressors (e.g. temporally and spatially varying recharge and groundwater
abstractions) on water table fluctuations. Naturally, this should also allow SWAT-MODFLOW to
provide more realistic assessments of the impacts of groundwater abstractions on streamflow in
comparison with SWAT.
While setting up the drinking water abstraction in SWAT, three limitations were identified, also
reported in (Molina-Navarro et al., 2019). The first is that SWAT only allows one decimal point for
abstraction numerical inputs with a unit of $10^4$ m$^3$ day$^{-1}$ for each month. This means that pumping rate
variations within one month cannot be simulated by SWAT and that the accuracy of abstraction
dynamics thus cannot be guaranteed. As a result of this limitation, the abstraction amount in SWAT
and SWAT-MODFLOW was not completely identical. The second limitation is that the abstraction
from deep aquifer did not result in any streamflow change. Therefore, all the abstraction sources had
to be defined as the shallow aquifer in SWAT to achieve a signal in streamflow despite that we had at
least three wells receiving water from a deep aquifer (the fifth layer according to the MODFLOW-
NWT set-up). The last limitation is that the abstraction rates of all wells in each subbasin in SWAT
have to be summed up to one input value, thereby ignoring the specific location of wells within
individual subbasins.



SWAT-MODFLOW overcomes the limitations in SWAT by exploiting the spatial explicitness of
MODFLOW where groundwater abstraction can be simulated using the Well Package, which allows
many decimal points for abstraction inputs as well as user-defined units, pumping rates at potentially
daily intervals, and wells located in any vertical layer and any grid cell within a subbasin. In addition
to the outputs from SWAT, SWAT-MODFLOW also provides fully distributed groundwater-related
outputs such as spatial-temporal patterns of water table elevation, distributed aquifer recharge, and
groundwater-surface water exchange rates at a cell level, permitting detailed analysis of groundwater
and its interaction with surface water. This may be an important input to groundwater resources
management (e.g. groundwater abstraction) and the solving of surface water rights issues. These
capabilities demonstrate the advantage of SWAT-MODFLOW over modifying the SWAT
groundwater module codes to improve groundwater flow simulation (Nguyen and Dietrich, 2018;
Pfannerstill et al., 2014; Zhang et al., 2016), which remains a semi-distributed way to simulate
subsurface hydrologic processes and does not generate detailed groundwater outputs. This point
supports the findings about the advantages of SWAT-MODFLOW over SWAT in (Molina-Navarro et
al., 2019) but using a much more complex set-up.
**4.3 Performance of SWAT-MODFLOW and SWAT relative to other recent studies**
In previous studies, after coupling a calibrated SWAT and calibrated MODFLOW model, the SWAT-
MODFLOW model complex was applied without further calibration (Bailey et al., 2016; Chunn et al.,
2019), with calibration against only streamflow observations (Molina-Navarro et al., 2019), with
separated calibration for streamflow and groundwater head (Guzman et al., 2015), or with simple
manual calibration by graphically comparing the simulated and observed streamflow and groundwater
head (Sith et al., 2019). Since both the SWAT and MODFLOW supporting software can use the inverse
modeling (IM) method for calibration, and parameter non-uniqueness is an inherent property of IM
(Abbaspour, 2015), the coupling of a calibrated SWAT and a calibrated MODFLOW cannot guarantee
a proper or sufficiently optimized parameter set for the integrated SWAT-MODFLOW model. Because
groundwater and surface water interact with each other, calibrating the simulation of one part does not
guarantee proper simulation of the other part. Application of a combined calibration approach based
on PEST allowed us to calibrate the SWAT-MODFLOW model by adjusting simultaneously SWAT
and MODFLOW parameters and using observations of both streamflow and groundwater table when
deriving the objective function. The calibration results demonstrated that the summary statistics of the
SWAT-MODFLOW performance were improved by this approach (Table 6).



The ability of SWAT-MODFLOW to evaluate the impacts of groundwater abstraction on streamflow
or groundwater-surface water interactions has been tested in the previous studies (Guzman et al., 2015;
Chunn et al., 2019; Molina-Navarro et al., 2019). (Molina-Navarro et al., 2019), for example, also
found that the SWAT model showed almost no impact of groundwater abstraction on streamflow
depletion. Besides due to the simple representation of groundwater dynamics, the other cause of this,
we believe, is that same as suggested above, that the impact of groundwater water removal by
abstractions on water table fluctuations is currently not accounted for in the groundwater discharge
calculation in the SWAT source code. Our findings are generally consistent with those of these
previous studies, although all of the studies tested the effects of groundwater abstraction only by
drinking water without considering irrigation and based on assumed drinking water pumping wells. In
addition, in all the previous SWAT-MODFLOW studies, the River Package in the MODFLOW model
was the only package used for simulating groundwater-surface water interaction, ignoring the potential
drain flow processes. The SWAT-MODFLOW complex used in our study was further developed to
allow application of the Drain Package and to allow also an auto-irrigation routine to extract water
from groundwater grid cells; in this way the impacts of groundwater abstraction for both drinking
water and irrigation could be assessed.
**4.4 Limitations and future research**
Several limitations to this study need to be acknowledged. The simulated head generated by the steady-
state model was used as the initial head conditions for the transient model, as also suggested in other
studies (Anderson et al., 2015; Doherty et al., 2010). The ideal simulated initial heads should be
calibrated with the observed initial heads. However, we did not have enough observed heads at the
beginning of the simulation period (1997), so we used the observed heads covering the period 1996-
2010 for calibrating the original steady-state MODFLOW-NWT to obtain the simulated initial heads.
Fortunately, the groundwater heads of the study area did not change much during the study period (Fig.
9, Fig. 10) and the difference inherently exists between the observed and simulated heads, indicating
that the error between the ideal simulated initial heads and the actually used simulated initial heads
was small.
An approach based on PEST was utilized to calibrate streamflow and groundwater table variation
simultaneously in our SWAT-MODFLOW simulation, which improved the model performance and
enabled parameter sensitivity analysis for the model. However, only two wells with relatively
continuous time series of observed groundwater head were available and used to calibrate the
groundwater variation. Ideally, calibration would involve more wells with continuous time series of



observed head, but this limitation is anticipated to be minor in our study as the groundwater head did
not change much in our simulations and the change mainly followed the variation of recharge with
precipitation as its source.
The average annual streamflow difference and the regular pattern of daily streamflow difference
between the abstraction scenarios and the no-wells scenario were generally explained well, but,
surprisingly and unexpectedly, the streamflow difference between the scenario with only drinking
water wells and the no-wells scenario on 24 March, 2010, simulated by SWAT-MODFLOW at two
stations, were positive, being 1.54 and 0.55 $m^3$ $s^{-1}$, respectively (Fig. 12). The streamflow difference
between the scenario with only irrigation wells and the no-wells scenario at station B on the extreme
peak flow day (16 October, 2014) simulated by SWAT was -5.2 $m^3$ $s^{-1}$ but then became positive next
day, which cannot be explained well to our best of knowledge so far. However, we found that the
general results of this study were not influenced when modifying the value of these two unexpected
points to be expected.
Both the SWAT and SWAT-MODFLOW simulations were based on the "best" parameter combination
achieved through calibration, which was deemed to be satisfactory for the purpose of this study.
However, complex models such as SWAT and SWAT-MODFLOW are subject to non-uniqueness (i.e.
more than one parameter combination may yield satisfactory results), so future studies may need to
consider the uncertainty due to, for example, parameter uncertainty. The calibration tool SWAT-CUP
has already been able to evaluate SWAT parameter uncertainty, whereas the new approach based on
PEST to calibrate SWAT-MODFLOW needs to be further explored to adapt for model uncertainty
analysis.
Our results support our original hypothesis that SWAT-MODFLOW can produce more reliable results
in the simulation of the effects of groundwater abstraction for either drinking water or irrigation on
streamflow patterns. In addition, SWAT-MODFLOW can produce more outputs than SWAT.
However, SWAT-MODFLOW also requires more effort and data to be set up and calibrated, and
longer time to run (around 6 hours for a 19-year simulation in SWAT-MODFLOW by a desktop with
an Intel® Core™ Processor i7-6700 CPU and 16 GB installed RAM versus 6 minutes for a SWAT
simulation). Therefore, the balance between scientific accuracy and the computational burden should
be defined relative to the study goal when choosing between SWAT and SWAT-MODFLOW in a
future study. But clearly, if the purpose of a study is to investigate effects of groundwater abstraction
on streams, the efforts should be focused on setting up and applying a fully-distributed model in
groundwater domain, such as SWAT-MODFLOW. A graphical user interface has also been developed





to couple SWAT and MODFLOW based on the publically available version of the SWAT-
MODFLOW complex (Park et al., 2018). Since the SWAT-MODFLOW complex used in this study
was newly developed and allowed use of the Drain Package and auto-irrigation, a new graphical user
interface based on the new SWAT-MODFLOW complex could ensure that a study such as that
presented here is repeated with less effort and technical challenges.
**5. Conclusions**
SWAT and SWAT-MODFLOW models with relatively complex set-ups were applied to a lowland
catchment, the Uggerby River Catchment in Northern Denmark. Model performance and the outcome
of four groundwater abstraction scenarios (with real wells and abstraction rates) were analyzed and
compared.
Generally both models simulated well the temporal patterns of streamflow at the two hydrological
stations during the calibration and validation periods. SWAT-MODFLOW, however, showed superior
performance when visualizing time series results and when comparing summary statistics.
Furthermore, SWAT-MODFLOW generates many additional outputs for groundwater analysis, such
as spatial-temporal patterns of water table elevation and groundwater-surface water exchange rates at
cell or subbasin level, improving water resources management in a groundwater-dominated catchment.
Abstraction scenarios simulated by SWAT and SWAT-MODFLOW showed different signals in
streamflow change. The simulations by both models indicated that drinking water abstraction caused
streamflow depletion and that irrigation abstraction caused a slight total flow increase (but decreased
the soil or aquifer water storage, which may influence the hydrology outside the catchment). However,
the impact of drinking water abstraction on streamflow depletion by SWAT was minimal and
underestimated, and the streamflow increase caused by irrigation abstraction was exaggerated
compared with the SWAT-MODFLOW simulation, which produced more realistic results.
Overall, the new SWAT-MODFLOW model calibrated by PEST, which included the Drain Package
and a new auto-irrigation routine, presented a better hydrological simulation, wider possibilities for
groundwater analysis, and more realistic assessments of the impact of groundwater abstractions (for
either irrigation or drinking water purposes) on streamflow compared with SWAT. Thus, SWAT-
MODFLOW can be used as a tool for managing water resources in groundwater-affected catchments,
taking into account its higher computational demand and more time consumption.



*Code and data availability.*
The land use map based on the Danish Area Information System is freely available from
(https://www.dmu.dk/1_viden/2_miljoe-tilstand/3_samfund/ais/3_Metadata/metadata_en.htm).
Climate data is available from the Danish Meteorological Institute (https://www.dmi.dk/). QGIS is
freely available from https://qgis.org/en/site/. QSWAT, SWATCUP, and the SWAT-MODFLOW as
well as its source codes are publicly available from https://swat.tamu.edu/software. The steady-state
MODFLOW set-up was provided by NIRAS upon request. The PEST utilities and tutorial are freely
downloadable from http://www.pesthomepage.org/Home.php. The source code, executable, and tutorial
for the further developed SWAT-MODFLOW are available on the SWAT website
(https://swat.tamu.edu/software/swat-modflow/). The two code files used for SWAT-MODFLOW
calibration by PEST will be available through repository on https://www.re3data.org/ when the paper
is accepted.

*Author contributions.* DT and WL designed the study. WL undertook all practical elements of the
study, including setting up and calibrating the models, analyzing results, producing figures, and writing
the manuscript. SP and RTB contributed the idea and developed the codes for use of the PEST
approach to calibrate the SWAT-MODFLOW model. RTB provided the knowledge to set up SWAT-
MODFLOW and further developed the SWAT-MODFLOW complex codes. DT, EMN, HEA, HT,
and AN provided most of the data and contributed with their knowledge to setting up and calibrating
the models. DT and EJ helped to analyze the results and contributed to the discussion. JSJ and JBJ
provided the original steady-state MODFLOW-NWT set-up and contributed with relative knowledge.
All co-authors contributed to the manuscript writing.

*Competing interests.* The authors declare that they have no conflict of interest.

*Acknowledgements.*
The first author was supported by grants from the China Scholarship Council. Erik Jeppesen and
Dennis Trolle were supported by the AU Centre for Water Technology (WATEC). We thank Chenda
Deng, Xiaolu Wei, and Zaichen Xiang for technique assistance and knowledge exchange during Wei
Liu's research stay at the Colorado State University. We also thank Anne Mette Poulsen for valuable
editorial comments.







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





**Table 1**. Farm types and crop rotations used to describe agricultural management in the Uggerby
River Catchment (W: winter, S: spring).

| Rotation type | Farm Type | Manure N (kg N/ha) | % Farm area | Rotation scheme | | | | |
|---|---|---|---|---|---|---|---|---|
| | | | | Year 1 | Year 2 | Year 3 | Year 4 | Year 5 |
| Agricultural land 1 | Mixed and horticulture | <50 | 31.0 | W. wheat | W. wheat | S. barley | W. rape | S. barley |
| Agricultural land 2 | Dairy/Cattle | 85-170 | 35.7 | S. barley | Grass | S. barley | Grass | Grass |
| Agricultural land 3 | Dairy/Cattle | 85-170 | 33.3 | S. barley | S. barley | W. wheat | Corn silage | Corn silage |






**Table 2**. Initial ranges and calibrated values of the selected parameters for SWAT calibration.

| Parameter | Description | Initial range | Calibrated values | |
|---|---|---|---|---|
| | | | Subbasins: 4,5,7-13 (upstream) | Subbasins: 1,3,6,14-19 (downstream) |
| v__SFTMP.bsn | Snowfall temperature (°C) | -1 – 1 | 0.175 | |
| v__SMFMN.bsn | Melt factor for snow on December 21 (mm $H_2O$ °C$^{-1}$ d$^{-1}$) | 1 – 2 | 1.287 | |
| v__SMFMX.bsn | Melt factor for snow on June 21 (mm $H_2O$ °C$^{-1}$ d$^{-1}$) | 1.6 – 3.5 | 2.467 | |
| v__SMTMP.bsn | Snow melt base temperature (°C) | -2.3 – 1 | -1.342 | |
| v__SURLAG.bsn | Surface runoff lag coefficient | 1 – 10 | 6.379 | |
| v__ALPHA_BF.gw | Baseflow alpha factor for shallow aquifer (l d$^{-1}$) | 0 – 1 | 0.453 | 0.639 |
| v__ALPHA_BF_D.gw | Baseflow alpha factor for deep aquifer (l d$^{-1}$) | 0 – 1 | 0.756 | 0.913 |
| v__ALPHA_BNK.rte | Baseflow alpha factor for bank storage (l d$^{-1}$) | 0 – 1 | 0.912 | 0.533 |
| v__CH_K2.rte | Effective hydraulic conductivity in main channel alluvium (mm h$^{-1}$) | 0 – 75 | 57.068 | 45.018 |
| r__CN2.mgt | Initial SCS runoff curve number for moisture condition II | -0.3 – 0.3 | -0.279 | 0.137 |
| r__DDRAIN.mgt | Depth to subsurface drain (mm) | -0.3 – 0.3 | 0.066 | -0.129 |
| v__EPCO.hru | Plant uptake compensation factor | 0.01 – 1 | 0.163 | 0.254 |
| v__ESCO.hru | Soil evaporation compensation factor | 0 – 1 | 0.466 | 0.931 |
| r__GDRAIN.mgt | Drain tile lag time (h) | -0.3 – 0.3 | 0.052 | -0.021 |
| v__GWQMN.gw | Threshold depth of water in the shallow aquifer required for return flow to occur (mm) | 0 – 2000 | 1435.04 | 960.32 |
| v__GW_DELAY.gw | Groundwater delay time (d) | 0 – 200 | 116.28 | 123.40 |
| v__GW_REVAP.gw | Groundwater "revap" coefficient | 0.02 – 0.1 | 0.092 | 0.0313 |
| r__OV_N.hru | Manning´s "n" value for overland flow | -0.2 – 0.2 | -0.037 | -0.025 |
| v__REVAPMN.gw | Threshold depth of water in the shallow aquifer for "revap" or percolation to the deep aquifer to occur (mm) | 1000 – 2000 | 1633.81 | 1521.80 |
| r__SOL_AWC().sol | Available water capacity of the soil layer (mm $H_2O$ mm soil$^{-1}$) | -0.8 – 0.8 | -0.674 | 0.786 |
| r__SOL_BD().sol | Moist bulk density (g cm$^{-3}$) | -0.2 – 0.2 | -0.067 | 0.156 |
| r__SOL_K().sol | Saturated hydraulic conductivity (mm h$^{-1}$) | -0.8 – 2 | 1.290 | 1.831 |
| r__TDRAIN.mgt | Time to drain soil to field capacity (h) | -0.3 – 0.3 | -0.097 | -0.210 |
| v__RCHRG_DP.gw | Deep aquifer percolation fraction | 0 – 0.4 | 0.296 | 0.219 |
| AUTO_WSTRS | Water stress threshold that triggers irrigation (mm) | 70 | 30, 40, 60 | |

Note: v_ means that the existing parameter value is to be replaced by a given value; r_ means that an existing parameter
value is multiplied by (1+ a given value).













**Table 3.** Initial values, ranges, and calibrated values of the selected parameters for SWAT-MODFLOW calibration using PEST.

| Parameter | Description | Initial value | Parameter ranges | Calibrated values |
|---|---|---|---|---|
| v__SFTMP.bsn | --- | 0.175 | -0.946–0.351 | 0.351 |
| v__SMFMN.bsn | --- | 1.287 | 1.117–1.424 | 1.424 |
| v__SMFMX.bsn | --- | 2.467 | 2.387–3.129 | 2.387 |
| v__SMTMP.bsn | --- | -1.342 | -1.687– -0.46 | -0.46 |
| v__SURLAG.bsn | --- | 6.379 | 4.452– 8.151 | 4.964 |
| v__ALPHA_BNK.rte[a] | --- | 0.912 | 0.7– 1 | 0.7 |
| v__ALPHA_BNK.rte[b] | --- | 0.533 | 0.206– 0.617 | 0.231 |
| v__CH_K2.rte[a] | --- | 57.068 | 29.322– 59.779 | 59.779 |
| v__CH_K2.rte[b] | --- | 45.018 | 30.246– 60.088 | 41.182 |
| r__CN2.mgt[a] | --- | -0.279 | -0.3– -0.106 | -0.3 |
| r__CN2.mgt[b] | --- | 0.137 | -0.019– 0.175 | 0.0004 |
| v__EPCO.hru[a] | --- | 0.163 | 0.077– 0.436 | 0.436 |
| v__EPCO.hru[b] | --- | 0.255 | 0.01– 0.334 | 0.304 |
| v__ESCO.hru[a] | --- | 0.466 | 0.227– 0.681 | 0.227 |
| v__ESCO.hru[b] | --- | 0.931 | 0.684– 1 | 0.943 |
| r__OV_N.hru[a] | --- | -0.037 | -0.2– -0.02 | -0.02 |
| r__OV_N.hru[b] | --- | -0.025 | -0.155– -0.005 | -0.023 |
| r__SOL_AWC().sol[a] | --- | -0.675 | -0.8– -0.316 | -0.508 |
| r__SOL_AWC().sol[b] | --- | 0.786 | 0.344– 0.8 | 0.8 |
| r__SOL_BD().sol[a] | --- | -0.067 | -0.187– -0.05 | -0.185 |
| r__SOL_BD().sol[b] | --- | 0.156 | 0.077– 0.2 | 0.172 |
| r__SOL_K().sol[a] | --- | 1.29 | 0.902– 2 | 0.902 |
| r__SOL_K().sol[b] | --- | 1.831 | 1.012– 2 | 1.012 |
| COND_1 | Drain conductance | 0.00467 | 0.00311 – 0.00622 | 0.00543 |
| COND_2 | Drain conductance | 0.02487 | 0.01658 – 0.03316 | 0.03316 |
| HK_CLAY | Hydraulic conductivity of clay (m s$^{-1}$) | 3.84E-08 | 1E-09 – 4.4E-08 | 2.2E-08 |
| HK_SILT | Hydraulic conductivity of silt (m s$^{-1}$) | 5.00E-07 | 1E-07 – 9E-07 | 1E-07 |
| HK_SS | Hydraulic conductivity of silty sand (m s$^{-1}$) | 6.70E-06 | 1.51E-06 – 7.50E-06 | 7.5E-06 |
| HK_SSCS | Hydraulic conductivity of silty sand and clean sand (m s$^{-1}$) | 1.79E-05 | 1E-05 – 8E-05 | 1.79E-05 |
| HK_CS | Hydraulic conductivity of clean sand (m s$^{-1}$) | 0.000327 | 1E-04– 5E-04 | 3.15E-04 |
| SS_CLAY | Specific storage of clay (m$^{-1}$) | 0.001099 | 9.19E-04 – 1.28E-03 | 1.28E-03 |
| SS_SILT | Specific storage of silt (m$^{-1}$) | 0.000755 | 4.92E-04 – 1.02E-03 | 1.02E-03 |
| SS_SAND | Specific storage of sand (m$^{-1}$) | 0.000166 | 1.28E-04 – 2.03E-04 | 2.03E-04 |
| SY_CLAY | Specific yield of clay (%) | 0.06 | 0.04 – 0.08 | 0.04 |
| SY_SILT | Specific yield of silt (%) | 0.2 | 0.15 – 0.25 | 0.22 |
| SY_SAND | Specific yield of sand (%) | 0.32 | 0.25 – 0.35 | 0.35 |
| AUTO_WSTRS | --- | 30, 40, 60 | 30, 40, 60, 80 | |

Notes: "a" means that the parameter applies to the upstream areas, including subbasins: 4, 5, 7-13, while "b" applies to downstream areas, including subbasins 1, 3, 6, 14-19. "---" indicates that the corresponding parameters can be found in Table 2.



1044          **Table 4**. The summary statistics for the calibrated MODFLOW performance.

| Layer number | The number of observed heads | $M_E$ (Mean error) | $M_{AE}$ (Mean absolute error) | $R_{MSE}$ (Root mean squared error) |
|---|---|---|---|---|
| Layer 1 | 453 | -0.59 | 1.94 | 2.84 |
| Layer 3 | 572 | -0.54 | 2.36 | 3.15 |
| Layer 5 | 38 | -1.24 | 3.44 | 5.00 |
| All | 1063 | -0.59 | 2.22 | 3.11 |




































**Table 5**. Performance statistics indices for daily runoff at the outlets of subbasin 13 and subbasin 18 during the calibration (2001-2008) and validation (2009-2015, in brackets) periods by SWAT, SWAT-MODFLOW without PEST calibration, and SWAT-MODFLOW with PEST calibration.

| Outlets | Used models | $P_{BIAS}$ | $N_{SE}$ | $R^2$ |
|---|---|---|---|---|
| Subbasin 13 outlet | SWAT | -3.9 (5.9) | 0.66 (0.50) | 0.67 (0.53) |
| | SWAT-MODFLOW without PEST calibration | -6.9 (1.7) | 0.72 (0.51) | 0.75 (0.60) |
| | SWAT-MODFLOW with PEST calibration | 1.9 (9.4) | 0.78 (0.54) | 0.78 (0.61) |
| Subbasin 18 outlet | SWAT | 2.0 (12.4) | 0.74 (0.47) | 0.74 (0.53) |
| | SWAT-MODFLOW without PEST calibration | 1.0 (11.0) | 0.77 (0.46) | 0.79 (0.57) |
| | SWAT-MODFLOW with PEST calibration | 3.3 (13.1) | 0.81 (0.53) | 0.82 (0.60) |






**Table 6**. Summary statistics for the SWAT-MODFLOW calibration result.

| Observation group | Number of observed data | Weight of observed data | Contribution to squared weighted residuals before calibration by PEST | Contribution to squared weighted residuals after calibration by PEST |
|---|---|---|---|---|
| Streamflow A | 2557 | 2 | 4410.3 | 3479.7 |
| Streamflow B | 2557 | 1 | 4911.7 | 4025.3 |
| Well A | 570 | 1 | 113 | 154.9 |
| Well B | 961 | 1 | 946.6 | 908.6 |
| Sum | 6645 | --- | 10381 | 8568.5 |






**Table 7**. Average annual summary of the main components in the hydrological cycle of the Uggerby
River Catchment during the study period (2002-2015) simulated by SWAT and SWAT-MODFLOW,
1161                  respectively.

| Components | SWAT | SWAT-MODFLOW |
|---|---|---|
| Precipitation (mm yr$^{-1}$) | 923 | 923 |
| Surface flow (mm yr$^{-1}$) | 22 | 30 |
| Lateral subsurface flow (mm yr$^{-1}$) | 89 | 64 |
| Tile drain flow (mm yr$^{-1}$) | 20 | 0 |
| Drain (MODFLOW, mm yr$^{-1}$) | 0 | 268 |
| Groundwater flow (mm yr$^{-1}$) | 257 | 22 |
| Total water yield (mm yr$^{-1}$) | 388 | 384 |
| Actual evapotranspiration (mm yr$^{-1}$) | 503 | 516 |
| Potential evapotranspiration (mm yr$^{-1}$) | 727 | 726 |
| Soil storage (mm yr$^{-1}$) | 32 | 22 |
| Average annual irrigation amount in the irrigated HRUs (mm yr$^{-1}$) | 137 | 133 |






**Table 8**. Average annual stream flow change (2002-2015) at subbasin 13 outlet and subbasin 18
outlet for each abstraction scenario from no-wells scenario and the corresponding annual abstraction
simulated in SWAT and SWAT-MODFLOW

| Scenarios | | Scenario 2 (Only drinking water wells) | | Scenario 3 (Only irrigation wells) | | Scenario 4 (Both drinking water and irrigation wells) | |
|---|---|---|---|---|---|---|---|
| Model | | SWAT | SWAT-MODFLOW | SWAT | SWAT-MODFLOW | SWAT | SWAT-MODFLOW |
| Average annual stream flow decrease(-) or increase(+) ($10^6$ m$^3$ yr$^{-1}$) | Subbasin 13 outlet | -0.024 | -1.10 | 0.61 | 0.24 | 0.59 | -0.73 |
| | Subbasin 18 outlet | -0.12 | -2.53 | 1.60 | -0.55 | 1.48 | -1.79 |
| Annual abstraction ($10^6$ m$^3$ yr$^{-1}$) | Subbasins 4-5, 7-13 | 1.10 | 1.28 | 17.86 | 19.45 | 18.96 | 20.73 |
| | The entire cathment excluding subbasin 19 | 4.01 | 3.96 | 40.54 | 39.26 | 44.55 | 43.22 |


Notes: Subbasin 13 outlet receives streamflow from subbasins 4-5, 7-13; Subbasin 18 outlet receives streamflow
from the entire catchment excluding subbasin 19.



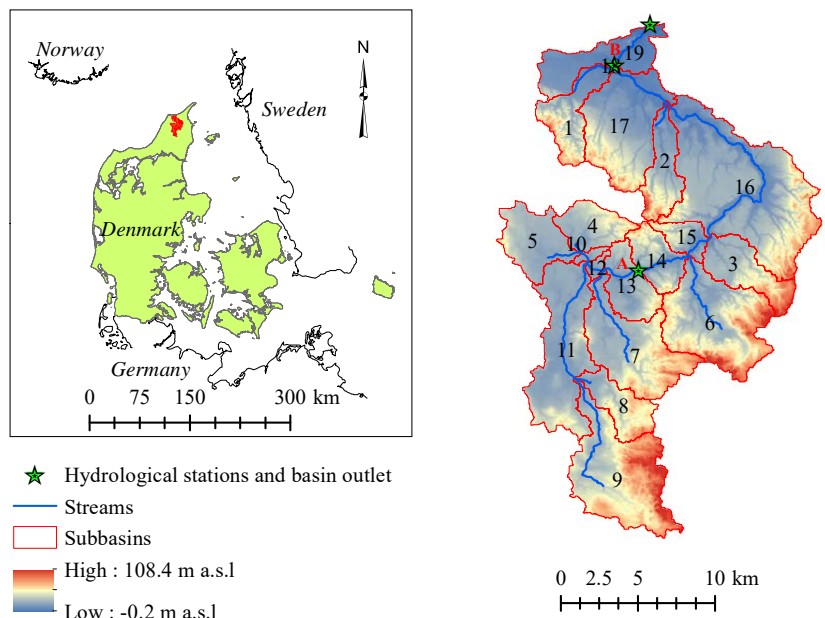

**Figure 1.** Location of the Uggerby catchment and its delineation in SWAT, including subbasins division, stream network definition based on the digital elevation model (DEM), hydrological monitoring stations, and basin outlet.

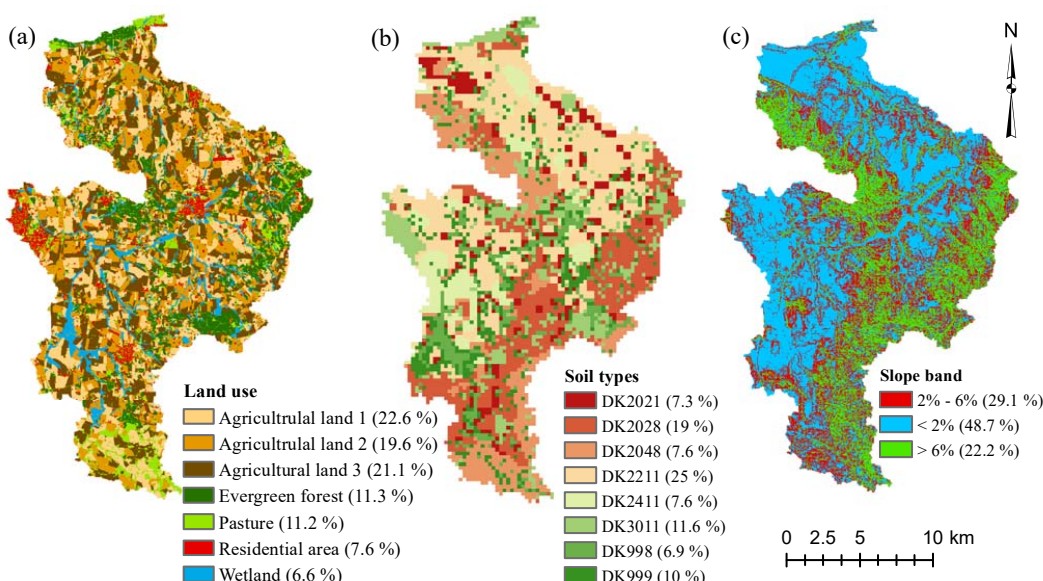

**Figure 2**. The distribution and proportion of each land use (a), soil type (b), and slope band (c) after reclassfication for HRU definition in SWAT.





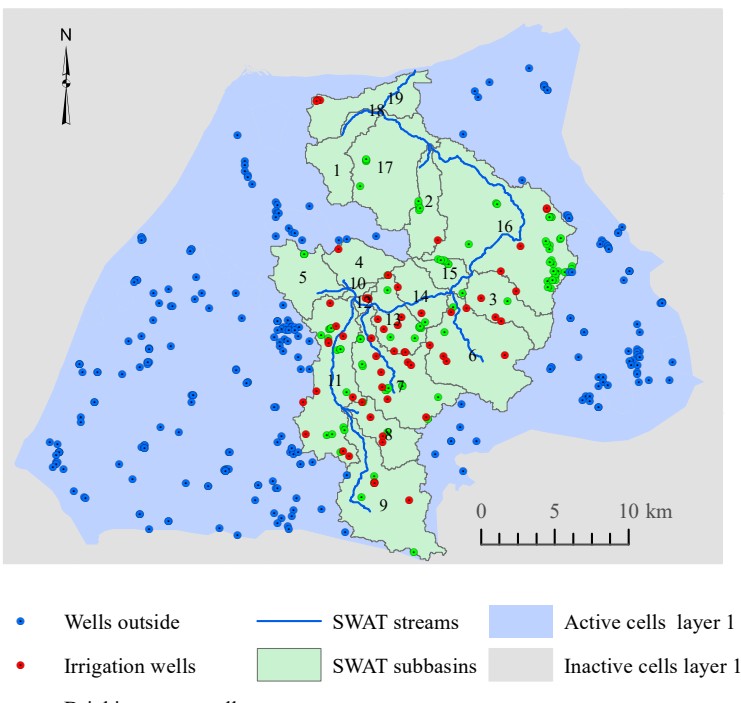

**Figure 3.** SWAT and MODFLOW set-up coverage and the well locations distributed inside or outside the Uggerby River Catchment.





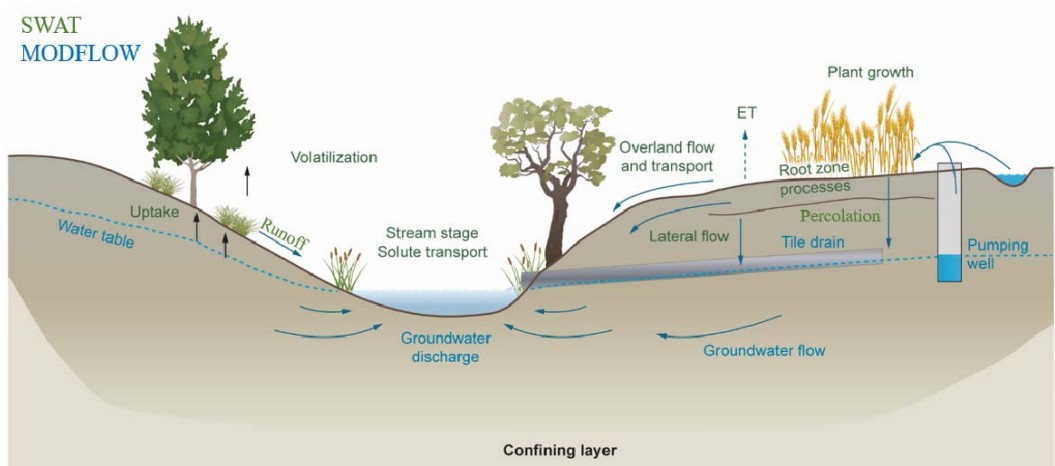

**Figure 4**. Schematic representation of water transport routes in stream-aquifer system as simulated by SWAT-MODFLOW, showing SWAT (green) and MODFLOW (blue) simulation processes. Adapted from (Molina-Navarro et al., 2019).



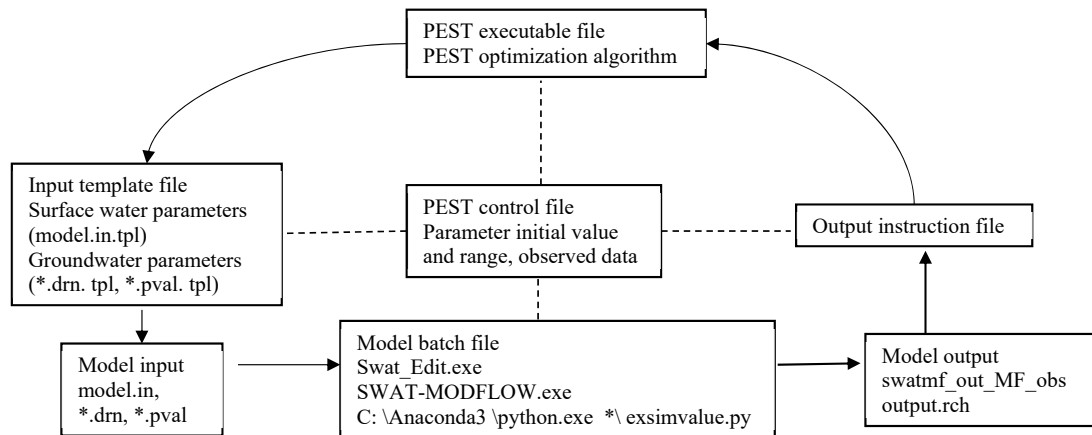

**Figure 5.** Schematic diagram of the PEST optimization process. The "*" means file name or file path.



**Figure 6**. The simulated and observed head contours as well as the locations of observed wells within layer 1 (a) and layer 3 (b), respectively.

**Figure 7.** Hydrographs of precipitation, observed and best simulated daily streamflow at the outlets of subbasin 13 (station A) and subbasin 18 (station B) during the calibration period (2002-2008) and the validation period (2009-2015) based on SWAT and SWAT-MODFLOW. The value in bracket is the discharge on 16 October, 2014, which is outside the range of the plot area.

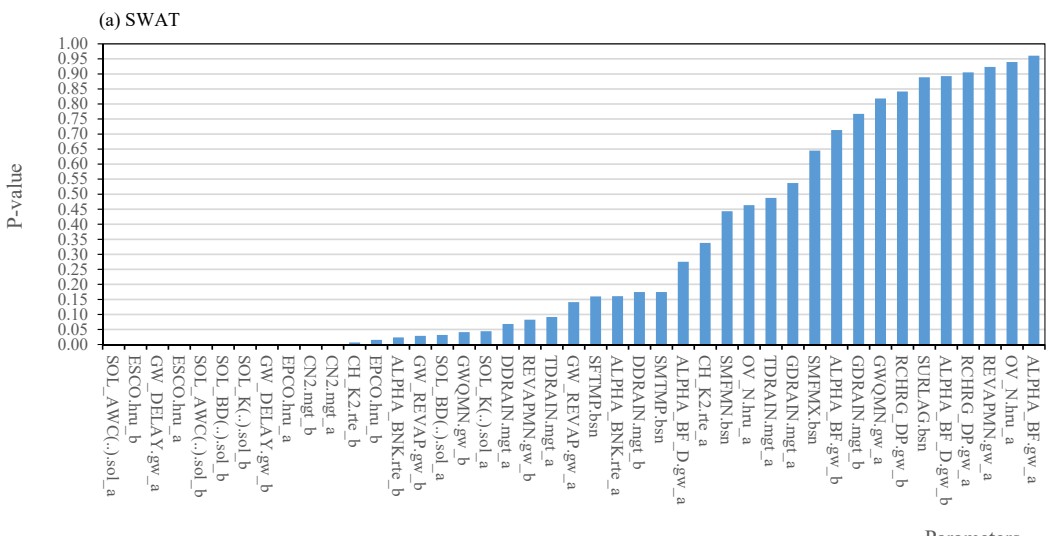

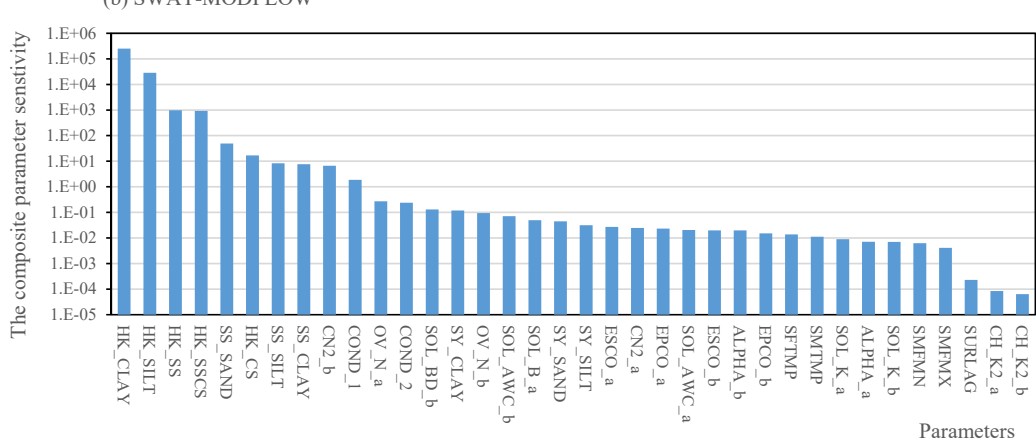

**Figure 8.** The sensitivity ranking of parameters in SWAT (a) and SWAT-MODFLOW (b) during calibration. The composite sensitivity of parameters was calculated based on the Jacobian matrix and the weight matrix after each PEST iteration and generated as an output once the PEST calibration was finished. The composite sensitivity values vary a little among the different iterations. The average value of each parameter among the 10 iterations for calibration is shown in the figure. More details regarding composite parameter sensitivity can be found in (Doherty, 2018). The "a" indicates that the parameter applies to the upstream areas, including subbasins: 4, 5, 7-13, and "b" indicates that the parameter applies to downstream areas, including subbasins 1, 3, 6, 14-19.



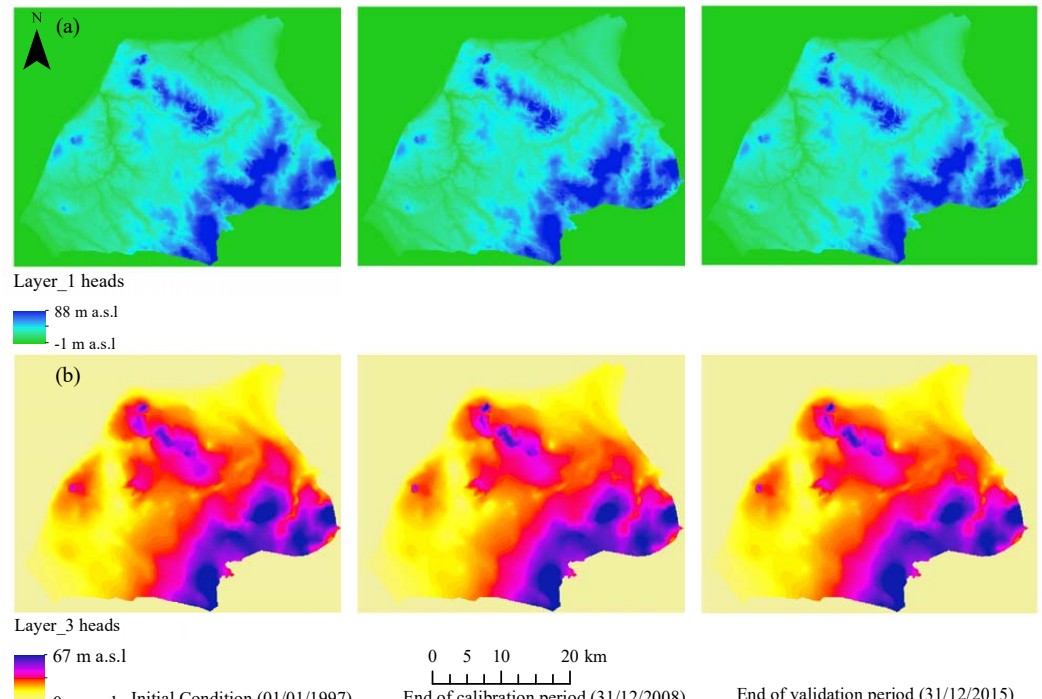

**Figure 9.** The simulated groundwater heads for the first layer (a) and third layer (b) at initial conditions, end of calibration period, and end of validation period by the calibrated SWAT-MODFLOW.



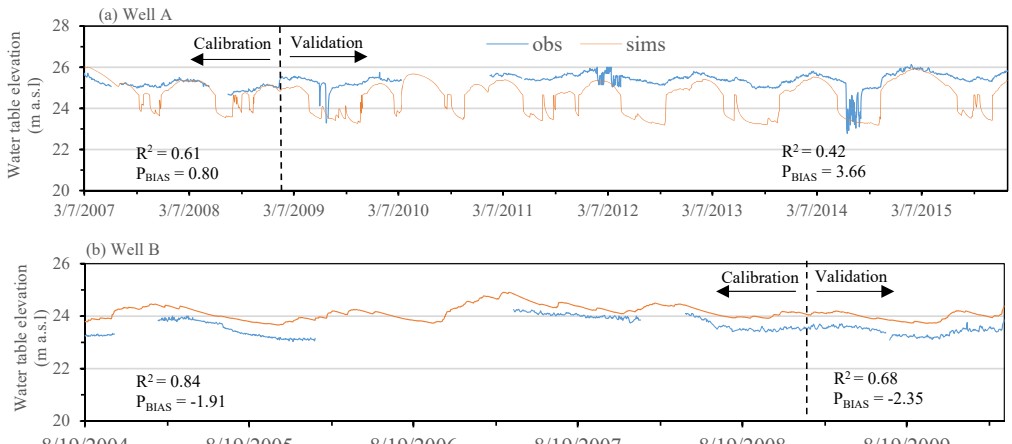

**Figure 10.** Hydrograph of daily simulated and observed groundwater heads (m a.s.l) of the two wells located in layer 1 used for calibrating the variation of groundwater heads simulated by SWAT-MODFLOW where relatively continuous observed data is available. Also shown are summary performance statistics.



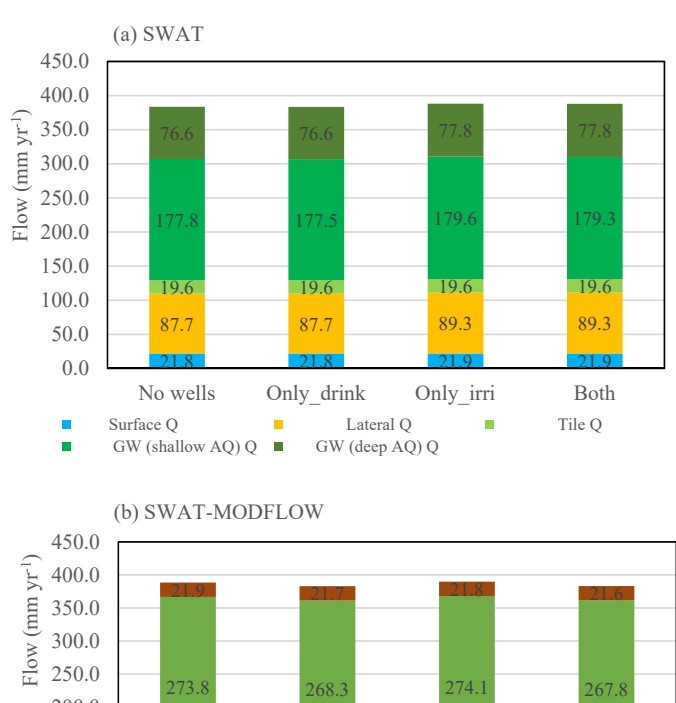

**Figure 11.** Average annual water yield (total flow) (2002-2015) simulated for the scenarios (no wells, scenario 1; only drinking water wells, scenario 2; only irrigation wells, scenario 3; both drinking water and irrigation wells, scenario 4) with SWAT (a) and SWAT-MODFLOW (b) and divided into flow components (Q = flow; GW = groundwater; AQ = aquifer).

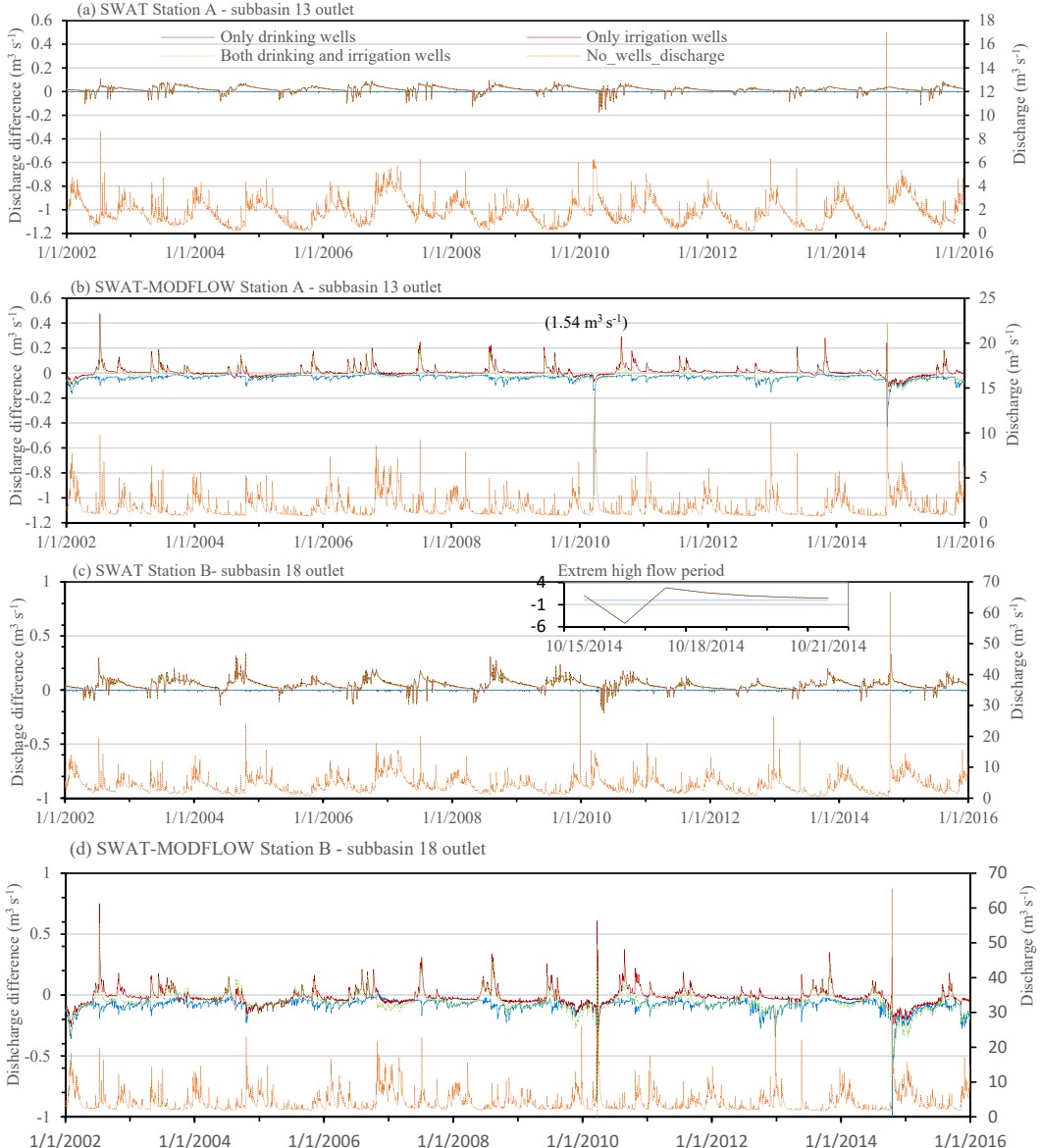

**Figure 12.** The simulated daily streamflow in the no-wells scenario and daily discharge differences between the abstraction scenarios (only drinking water wells, scenario 2; only irrigation wells, scenario 3; both drinking water and irrigation wells, scenario 4) and the no-wells scenario (scenario 1) at the outlets of subbasin 13 (station A) and subbasin 18 (station B) during the entire study period (2002-2015) based on SWAT and SWAT-MODFLOW, respectively. The value 1.54 m$^3$ s$^{-1}$ in brackets is the streamflow difference between the no-wells scenario and the scenario with only drinking water wells on 24 March, 2010, which is outside the range of the plot area.





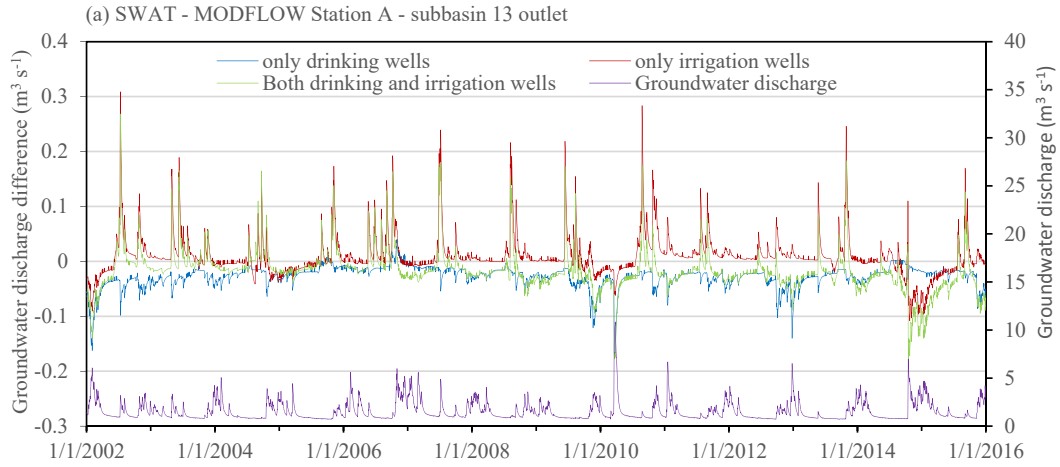

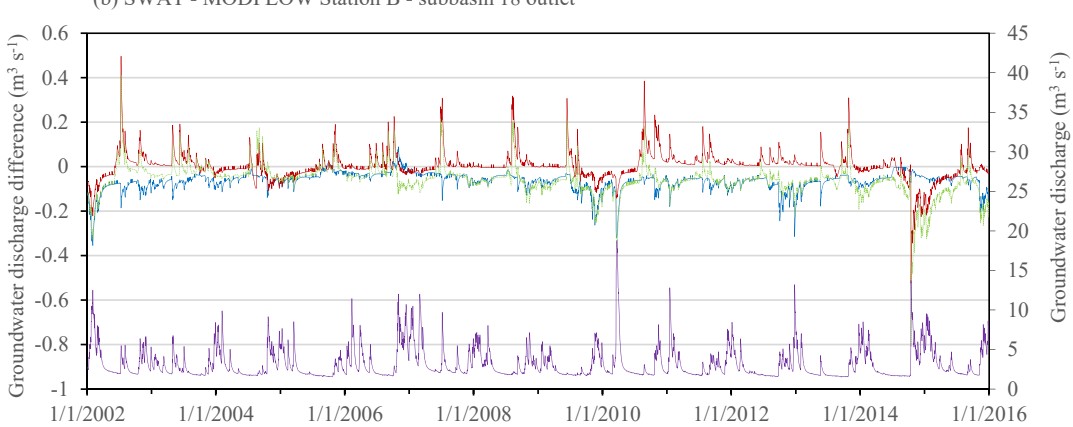

**Figure 13**. The hydrograph of simulated daily groundwater discharge to the stream network in the no-wells scenario and daily groundwater discharge differences between the abstraction scenarios (only drinking water wells, scenario 2; only irrigation wells, scenario 3; both drinking water and irrigation wells, scenario 4) and the no-wells scenario (scenario 1) in the upstream area of station A (a) and upstream area of station B (b), respectively, during the entire study period (2002-2015), based on SWAT-MODFLOW.