# Peer review of "Comparing SWAT with SWAT-MODFLOW hydrological simulations when assessing"

_Hydrology and Earth System Sciences, 2019_

## Referee Comment (RC1) · Anonymous Referee #1 · 2 Aug 2019

The main objective of the manuscript "Comparing SWAT with SWAT-MODFLOW hydrological simulations when assessing the impacts of groundwater abstractions for irrigation and drinking water" was to evaluate watershed simulations between the SWAT and the coupled SWAT-MODFLOW models in which groundwater extractions are important. Specific objectives were to incorporate the MODFLOW Drain package, SWAT auto-irrigation, and a coupled calibration capability using PEST within the SWAT-MODFLOW modeling framework. The authors concluded that the SWAT-MODFLOW model calibrated by PEST shown better performance when compared with SWAT.

[Figure]

Aside from incorporating PEST to calibrate the coupled model the manuscript does not support additional novelty. Note that in the publication "A model integration framework for linking SWAT and MODFLOW" the MODFLOW drainage package, SWAT autoirrigation, and the MODFLOW NWT was already integrated into the SWAT-MODFLOW framework. However, using PEST to simultaneously calibrate both models may be of interest for the SWAT modelers community. The introduction needs extensive revision. I recommend that the authors refocus and simplify the objectives of the manuscript.

As the extent of both watersheds were different, how do the authors estimated the percolating fluxes in the groundwater extent that were not simulated by SWAT?

Line 74: Are the authors refereeing to conceptual models? What is an analytical model? I think SWAT as well as MODFLOW fall in the category of analytical models.

Line 79: The authors should be more precise in the model terminology when referring to hydrologic models. This is somewhat confusing.

Lines 120-159: The authors should revise these two paragraphs in a way that clearly illustrate readers the difficulties of coupling a semi-distributed surface model (SWAT) with a fully distributed groundwater model (MODFLOW). In addition, the authors should help the reader to realize that the spatial discretization that results from SWAT HRUs result in clustering hydrological processes that are geographically disconnected but grouped by slope, land use and "soil type", as an example of this are fundamental processes such as evapotranspiration and irrigation. Note that irrigation is applied to the HRU level regardless if the area that is represented is contiguous o spatially disconnected and so, infiltration fluxes. This is a major model structure uncertainty in the SWAT model that pass unnoticed when simulated surface processes but problematic when integrating or coupling it with groundwater fully distributed model.

Line 137: I think that the correct sentence must acknowledge that the mentioned authors already integrated or coupled SWAT and MODFLOW instead of they try.

Lines 160-174: The full story is interesting but the claim that SWAT may a suitable model to represent these hydrological features with accuracy for comparison is non-sense. The coupled SWAT-MODFLOW model may be a better approach that is able to capture this surface-subsurface interaction but the MODFLOW spatial discretization of 100x100m is quite coarse for these complex surface-subsurface water interactions. I recommend the authors to focus on illustrating the weakness and advantages of the NIRAS MODFLOW model and where the coupled version may be able to advance this model.

2.1. Study Area Is there water intrusion and lunar tidal influence at the watershed outlet? I recommend to include the groundwater watershed in this section

228-232: Can the author expand or provide a short explanation in regards these gridded datasets. Is this data the result of spatial interpolation from ground-based stations or there is a gridded network of stations? Also, if there is an additional step to generate the gridded datasets, can the authors illustrate the limitations and report the methodology behind it?

247: Is this not the opposite. Please correct me if I am wrong, the Well package in MODFLOW need the extraction volumes' as an input data which it was estimated by the SWAT auto-irrigation module.

248: Where did the authors collect the drinking water data?

254: Can the authors provide a figure that illustrates the hydrogeology of the study area?

398: I don't find this a large number of cell

399-401: I am not sure if this is relevant here. If the manuscript was intended for algorithms and cyber optimization it may be relevant.

411-427: I think here is where the manuscript totally misses the focus.

657-659: This is not true, in fact, the authors will have a hard time demonstrating this. What is important is here is the proper representation in space and time of the hydrological processes occurring in the two domains (water circulation), for instance, feedback fluxes from the groundwater domain, infiltration fluxes from irrigated areas, and the correctness of estimated extracted volumes by the auto-irrigation module. In addition to this, the constraints set by the modelers, and the spatial discretization of the input data.

667: groundwater in SWAT is referred to as the processes occurring to the 6m soil dept. This is just confusing in the SWAT literature. On the other hand, in SWAT the deep percolating water and the deep aquifer is what in MODFLOW is referred to groundwater processes which SWAT model it as a loss in the system.

285-292: From what I had read to this point, I will say that this may be the contribution of this manuscript as all the rest is already published except PEST to calibrate both models simultaneously. Why no focusing in demonstrating that groundwater feedback fluxes were incorporated in the model and how irrigation was narrowed in the fuzzy HRU-MODFLOW cell definition?

301: I don't get this. . . why SWAT-cup, when the authors are claiming the use of PEST for calibration?

Figure 3. I recommend replacing "coverage" by domain or extent.

Figure 4. This figure does not properly represent the process representation of the SWAT model. Note that the SWAT model is semi-distributed and remains semi-distributed even though it is coupled with MODFLOW.

Figure 5. The authors MUST acknowledge that this figure was modified from Zhulu Lin document (https://www.ndsu.edu/pubweb/~zhulin/pdf/teaching/starting%20pest.pdf)

Figure 8. Please rank the x-axis in figures (a) and (b) in identical order. This will facilitate comparison.

Figure 9. This figure is useless. If the authors' fid that there is something that is necessary to communicate to the reader, they need to find a different way to illustrate the changes.

Figure 10. It seems like simulations in Well A poorly represent the dynamic. Well B seems to have a systematic bias.

The authors should include some model performance metrics when comparing the models

---

## Referee Comment (RC2) · Anonymous Referee #2 · 9 Aug 2019

Liu et al. present comparison of SWAT with SWAT-MODFLOW for the Uggerby catchment, Denmark. The study is well written with excellent level of detail provided in the method. I have no working knowledge of the models applied in this study, so my comments are high level, relating to the statistical interpretation of the results and their significance.

My main concern is that the conclusion that SWAT-MODFLOW is superior does not seem to be justified by model performances achieved in validation. Looking at Table 5, we see that the additional model flexibility offered by MODFLOW and MODFLOW

with PEST significantly increase calibration performance without improving validation performance significantly. This suggests that these models are simply over-fitting. I don't think this paper requires any additional experiments to be run, but I do suggest that the results need to be interpreted accurately. Unless the authors offer a convincing reason their current interpretation of the the results (based on their validation) I would suggest that the whole discussion and conclusions need to be rewritten to be more reflective of an honest appraisal of the model performances.

The other general issue is that the paper is very long. I think it can be shortened significantly without losing the key messages.

Specific comments:

Abstract - very long; considering shortening. Line 80 - no model considers the "entire" complexity. Please revise. Line 160 - please report % of crop production dependent on irrigation so the reader can get a sense of the importance. Line 173 - The hypothesis that the "benefits of applying SWAT-MODFLOW outweigh the costs" is one that can be tested objectively (and is not answered in your results). I suggest reframing the study so that the aim is to explore effects of introducing MODFLOW and MODFLOW pest into SWAT simulations in this particular catchment. Line 325 - Is the water stress threshold taken as a single value for the whole catchment? If so, what are the limitations of this assumption? Would the threshold vary according to crop type / soil type? Page 12 - the reader does not need to know the names of your python scripts. Figure 5 - not particularly helpful. I think this can be omitted. Line 478 (and throughout the results section)... lots of results reported in vague terms ("little higher", "much lower" ... etc). Please report % change instead. Line 552 - it does not reflect a shortcoming of SWAT groundwater module if the improvements are simply overfitting. Line 560 - this conclusion is not warranted if the model has been overfitted, as is suggested by results reported in Table 5. Figure 6 - remove background shading. Figure 10 - These are not promising results. Seasonal well drawdowns in the simulations are do not occur in the observations. Why should this not be reported as evidence of poor performance of

SWAT-MODFLOW? Lastly, the arbitrary labels attached to NSE scores ("satisfactory" etc) are inappropriate. Report the numbers, show the data, and let the reader decide what is satisfactory.

---

## Author Comment (AC1) · 17 Oct 2019

We are grateful to Anonymous Referee 1 for her/his time and effort and for providing constructive comments.

**General comments**: The main objective of the manuscript "Comparing SWAT with SWAT-MODFLOW hydrological simulations when assessing the impacts of groundwater abstractions for irrigation and drinking water" was to evaluate watershed simulations between the SWAT and the coupled SWAT-MODFLOW models in which groundwater

extractions are important. Specific objectives were to incorporate the MODFLOW Drain package, SWAT auto-irrigation, and a coupled calibration capability using PEST within the SWAT-MODFLOW modeling framework. The authors concluded that the SWAT-MODFLOW model calibrated by PEST shown better performance when compared with SWAT.

Aside from incorporating PEST to calibrate the coupled model, the manuscript does not support additional novelty. Note that in the publication "A model integration framework for linking SWAT and MODFLOW" the MODFLOW drainage package, SWAT auto-irrigation, and the MODFLOW-NWT were already integrated into the SWAT-MODFLOW framework. However, using PEST to simultaneously calibrate both models may be of interest to the SWAT modeler's community. The introduction needs extensive revision. I recommend that the authors refocus and simplify the objectives of the manuscript.

**Response:** We thank the reviewer for the comments made. The reviewer has raised a valid point about highlighting the key innovations of our study. We acknowledge that the MODFLOW drainage package, SWAT auto-irrigation, and the MODFLOW-NWT were integrated into the SWAT-MODFLOW framework in the publication "A model integration framework for linking SWAT and MODFLOW" by Guzman et al. (2015). The codes of the SWAT-MODFLOW framework outlined in that publication are, however, not publically available for further development or for scientific review. In our study, to enable the application of the Drain Package and an auto-irrigation routine, we further developed the open-source SWAT-MODFLOW complex based on the previous version (v2) developed by Bailey et al. (2016), for which the codes are publically available. Besides public availability, this edition also has some additional advantages over previous coupled SWAT-MODFLOW frameworks: an efficient HRU-grid cell mapping scheme with the generation of geographically explicit HRUs, the ability to use SWAT and MODFLOW models of different spatial domains, and a graphical user interface available for facilitating its application. It has been applied to many catchments of varying sizes worldwide, as mentioned in the introduction of our manuscript. Further development

of this publically available edition would benefit the SWAT-MODFLOW user community. To accommodate the reviewer, we have now clarified that we have further developed the SWAT-MODFLOW complex based on the publically available version originally by Bailey et al. (2016).

Modifications: Line 21: "The SWAT-MODFLOW complex was further developed to enable the application of the Drain Package and an auto-irrigation routine based on the previous publically available version."

Line 176-179: "The SWAT-MODFLOW complex used in this study was further developed based on the publically available version (https://swat.tamu.edu/software/swat-modflow/) to enable the application of the Drain Package of MODFLOW and to allow auto-irrigation."

Lin 662-665: "In addition, in all the previous studies using the SWAT-MODFLOW developed by Bailey et al. (2016), the River Package in the MODFLOW model was the only package used for simulating groundwater-surface water interaction, ignoring the potential drain flow processes."

**Specific comments:**

**1)** As the extent of both watersheds were different, how did the authors estimate the percolating fluxes in the groundwater extent that were not simulated by SWAT?

**Response:** Only the domain covered by both SWAT and MODFLOW was coupled, and the original functionality of MODFLOW or SWAT was retained beyond the common domain. The original MODFLOW from NIRAS is steady-state and NIRAS used the average net precipitation from the national DK-Model (www.vandmodel.dk) from 1999 to 2008 as its percolating fluxes (recharge).

Modifications: We have added the following sentences into the manuscript:

Line 151-152: "Only the domain covered by both SWAT and MODFLOW was coupled, and the original functionality of MODFLOW or SWAT was retained beyond the common

domain."

Line 265-267: "The average net precipitation from the national DK-Model (www.vandmodel.dk) from 1999 to 2008 was used as the recharge data in the Recharge Package."

**2)** Line 74: Are the authors refereeing to conceptual models? What is an analytical model? I think SWAT as well as MODFLOW fall in the category of analytical models.

**Response:** We are refereeing to numerical models. In mathematics, a problem can be solved either analytically or numerically. An analytical solution produces exact results, while a numerical solution approximates the solution through discrete numerical (time) steps. MODFLOW is a numerical model.

**3)** Line 79: The authors should be more precise in the model terminology when referring to hydrologic models. This is somewhat confusing.

**Response:** We have modified the sentence in the first part of line 79. As for the second half of line 79, MODFLOW is a numerical and process-based model.

Modifications: Line 76: "Nevertheless, as they do not simulate many of the physical processes and ignore the real-world complexity, they may render results far away from reality."

**4)** Lines 120-159: The authors should revise these two paragraphs in a way that clearly illustrates readers the difficulties of coupling a semi-distributed surface model (SWAT) with a fully distributed groundwater model (MODFLOW). In addition, the authors should help the reader to realize that the spatial discretization that results from SWAT HRUs result in clustering hydrological processes that are geographically disconnected but grouped by slope, land use and "soil type", as an example of this are fundamental processes such as evapotranspiration and irrigation. Note that irrigation is applied to the HRU level regardless if the area that is represented is contiguous or spatially disconnected and so, infiltration fluxes. This is a major model structure uncertainty in the

SWAT model that passes unnoticed when simulated surface processes but problematic when integrating or coupling it with groundwater fully distributed model.

**Response:** Guzman et al. (2015) and Bailey et al. (2016) have illustrated the framework development of SWAT-MODFLOW and the difficulties during the coupling in detail. The main goal of our study was not to illustrate the framework development of SWAT-MODFLOW once again, but rather to apply the model for an actual case. However, the reviewer has raised an important point about the HRUs.

Modifications: We have rephrased the text in the lines 105-107 as follows:

Lines 103-105: "HRUs are modelled as lumped and non-geo-located within each sub-basin (Guzman et al., 2015), which makes SWAT computationally efficient for long-term simulation, but this comes with the sacrifice of spatial discretization of HRUs."

**5)** Line 137: I think that the correct sentence must acknowledge that the mentioned authors already integrated or coupled SWAT and MODFLOW instead of they try.

**Response:** Good point.

Modifications: We have modified the sentence in line 137 as follows:

Line 135-136: "There are a few studies that have integrated SWAT and MODFLOW code into one model complex (Kim et al., 2008; Yi and Sophocleous, 2011; Guzman et al., 2015; Bailey et al., 2016)."

**6)** Lines 160-174: The full story is interesting but the claim that SWAT may be a suitable model to represent these hydrological features with accuracy for comparison is nonsense. The coupled SWAT-MODFLOW model may be a better approach that is able to capture this surface-subsurface interaction but the MODFLOW spatial discretization of 100x100m is quite coarse for these complex surface-subsurface water interactions. I recommend the authors to focus on illustrating the weakness and advantages of the NIRAS MODFLOW model and where the coupled version may be able to advance this model.

**Response:** With the simplified implementation of groundwater dynamics in SWAT, the SWAT model can mislead the evaluation of groundwater resources or perform rather poorly in catchments where the streamflow is strongly dependent on groundwater discharge (Gassman et al., 2014). Therefore, it would be incorrect to state that SWAT is a suitable model for representing the hydrological features with accuracy, especially in groundwater-dominated catchments. The reviewer has raised a valid point about how the coupled SWAT-MODFLOW advance the MODFLOW (e.g. through spatially explicit recharge). This is not a focal point of our study, but this aspect is introduced in the third paragraph of the introduction.

**7)** 2.1. Study Area. Is there water intrusion and lunar tidal influence at the watershed outlet? I recommend to include the groundwater watershed in this section.

**Response:** As far as we know the influence of water intrusion and lunar tidal on the outflow of the watershed is small. It is a good suggestion that the groundwater watershed should be included in the study area section.

Modifications: We have added the groundwater domain and wells to the map of the study area (Fig.1, as attached) and removed the figure 3. We also have removed the sentences related to Figure 3 in line 411-415. The caption of Fig.1 has been modified into "Location of the Uggerby River catchment and Hjørring Municipality, and their delineation in SWAT and MODFLOW. The locations of wells distributed inside or outside the Uggerby River Catchment are also shown."

The sentence in line 202-204 has been rephrased as follows:

Line 201-204: "According to an investigation carried out by Hjørring Municipality in 2009, 101 drinking water pumping wells and 57 irrigation pumping wells placed on pasture and agricultural land were registered within the Uggerby River catchment, and another 256 wells exist outside the catchment but inside Hjørring Municipality (Fig. 1)."

The sentence in line 189-191 has been simplified as follows:

Line 189-190: "The Uggerby River originates from the southern part of Hjørring and discharges into the coast of the North Sea."

**8)** 228-232: Can the author expand or provide a short explanation in regards to these gridded datasets. Is this data the result of spatial interpolation from ground-based stations or there is a gridded network of stations? Also, if there is an additional step to generate the gridded datasets, can the authors illustrate the limitations and report the methodology behind it?

**Response:** The Danish Meteorological Institute provides gridded data based on spatial interpolation from ground-based stations throughout Denmark. This is described in more detail in (Lu et al., 2016), which we also refer to.

**9)** 247: Is this not the opposite. Please correct me if I am wrong, the Well package in MODFLOW need the extraction volumes' as an input data which it was estimated by the SWAT auto-irrigation module.

**Response:** The auto-irrigation routine in the SWAT-MODFLOW of this study was designed as the reviewer described. However, the sentence in line 247 refers to the set-up of drinking water wells in SWAT. For comparison of the two model's performance on hydrology, we need to make the groundwater abstraction amount in the SWAT and SWAT-MODFLOW model set-ups approximately equivalent. Therefore, we set up the drinking water wells in SWAT according to the Well package extractions in MODFLOW.

Modifications: We have rephrased the sentence in lines 247-249 as follows to make the meaning more clear:

Lines 246-249: "With the number and location of pumping wells as well as their pumping rates obtained from the Well Package in the MODFLOW model, the water abstraction amounts from drinking water wells were added up in each subbasin and set as the water use pumped from the shallow aquifer in SWAT."

**10)** 248: Where did the authors collect the drinking water data?

**Response:** The locations of the drinking water wells are recorded in the national database. As Denmark regulates the groundwater abstraction through implementing a permit authority system of groundwater abstractions, the regulation (abstraction) levels were assigned as the pumping rates in the MODFLOW set-up.

**11)** 254: Can the authors provide a figure that illustrates the hydrogeology of the study area?

**Response:** Yes.

Modifications: We have plotted a figure showing the hydraulic conductivities of each cell in each layer in the steady-state MODFLOW-NWT set-up and have added it to the Appendix as appendix A, as attached below. We also have added the following sentence into the manuscript:

Line 261-263. "The first, third and fifth layers are dominated by sand with relatively large hydraulic conductivities, while the second and fourth layers are dominated by clay with lower hydraulic conductivities (Appendix A)."

**12)** 398: I do not find this a large number of cells.

**Response:** We have now deleted the sentence in lines 399-401 to shorten the text.

**13)** 399-401: I am not sure if this is relevant here. If the manuscript was intended for algorithms and cyber optimization, it may be relevant.

**Response:** We have deleted the sentence in lines 399-401 to shorten the text.

**14)** 411-427: I think here is where the manuscript totally misses the focus.

**Response:** Actually, the main objective of the manuscript was to compare SWAT with SWAT-MODFLOW hydrological simulations when assessing the impacts of groundwater abstractions for irrigation and drinking water, as the title also indicates.

[Figure]

Modifications: We have removed the sentences in line 411- 415 to shorten the texts.

**15)** 657-659: This is not true, in fact, the authors will have a hard time demonstrating this. What is important is here is the proper representation in space and time of the hydrological processes occurring in the two domains (water circulation), for instance, feedback fluxes from the groundwater domain, infiltration fluxes from irrigated areas, and the correctness of estimated extracted volumes by the auto-irrigation module. In addition to this, the constraints set by the modelers, and the spatial discretization of the input data.

**Response:** The reviewer has raised a valid point. Ideal calibration of the model involves more data, such as feedback fluxes and infiltration fluxes. Unfortunately, we do not have these data for ideal calibration. Modifications: We have now rephrased the text in lines 659-662 as follows:

Lines 650-654: "Application of a combined calibration approach based on PEST allowed us to calibrate the SWAT-MODFLOW model by adjusting simultaneously SWAT and MODFLOW parameters and against observations of both streamflow and groundwater table, though a more ideal calibration involves more observed data, such as feedback fluxes from the groundwater domain, infiltration fluxes in irrigated areas."

**16)** 667: Groundwater in SWAT is referred to the processes occurring to the 6m soil dept. This is just confusing in the SWAT literature. On the other hand, in SWAT the deep percolating water and the deep aquifer is what in MODFLOW is referred to groundwater processes which SWAT model it as a loss in the system.

**Response:** Actually, only much older SWAT versions considered the deep percolating water as a loss in the system, as described in the SWAT theory book published in 2009 (Neitsch et al., 2011). In the updated SWAT versions (since around 3 years ago), percolation water to both shallow and deep aquifer may return and contribute to streamflow as baseflow.

**17)** 285-292: From what I had read to this point, I will say that this may be the contribution of this manuscript as all the rest is already published except PEST to calibrate both models simultaneously. Why no focusing on demonstrating that groundwater feedback fluxes were incorporated in the model and how irrigation was narrowed in the fuzzy HRU-MODFLOW cell definition?

**Response:** All innovations about the further development of SWAT-MODFLOW in this study were based on the previous publicly available version developed by Bailey et al. (2016). Since Bailey et al. (2016) has already illustrated the two-way interactions between surface water and groundwater and how HRU-MODFLOW cells are connected in detail, we do not intend to duplicate their efforts, but rather we refer to their detailed paper.

**18)** 301: I don't get this. . . why SWAT-cup, when the authors are claiming the use of PEST for calibration?

**Response:** We used SWAT-CUP to calibrate SWAT. Since SWAT-CUP can only be used for calibrating the SWAT parameters, we developed and utilized the PEST approach with some files from SWAT-CUP to calibrate SWAT and MODFLOW parameters simultaneously in SWAT-MODFLOW against the observations of both streamflow and groundwater table.

Modifications: We have rephrased the text in lines 342-345 as follows:

Lines 344-349: "Since SWAT-CUP can only be used to calibrate SWAT parameters, the PEST approach was developed and utilized to adjust SWAT and MODFLOW parameters simultaneously. However, SWAT-MODFLOW can also be run through SWAT-CUP, whereby the summary statistics of model performance can be derived and directly compared between SWAT and SWAT-MODFLOW. In addition, model.in and Swat Edit.exe, which are included in the creation of the SWAT-CUP project folder, were used to adjust SWAT parameters within the PEST routine."
**19)** Figure 3. I recommend replacing "coverage" by domain or extent.

**Response:** We have combined Figure 1 and Figure 3 into one figure, as attached below, and have deleted the original caption of Figure 3.

**20)** Figure 4. This figure does not properly represent the process representation of the SWAT model. Note that the SWAT model is semi-distributed and remains semi-distributed even though it is coupled with MODFLOW.

**Response:** We agree that the SWAT model remains semi-distributed after being coupled with MODFLOW. Bailey et al. (2016) have illustrated this with a figure in detail.

Modifications: We have added the following sentence into the caption of Figure 4: "After being coupled with MODFLOW, the overland part of the SWAT model remains semi-distributed, while the HRU-calculated percolation from SWAT model is explicitly spatial."

**21)** Figure 5. The authors MUST acknowledge that this figure was modified from Zhulu Lin document (https://www.ndsu.edu/pubweb/âĹijzhulin/pdf/teaching/starting

**Response:** We have deleted Figure 5, as suggested by the other reviewer.

**22)** Figure 8. Please rank the x-axis in figures (a) and (b) in identical order. This will facilitate comparison.

**Response:** We used P-value and the composite parameter sensitivity, respectively, to rank the sensitivities of parameters. The smaller the P-value, the more sensitive the parameter. However, the larger the composite parameter, the more sensitive the parameter. For facilitating the comparison, we ranked the parameters according to their sensitivities from highest to lowest, as shown in Figure 8.

**23)** Figure 9. This figure is useless. If the authors find that there is something that is necessary to communicate to the reader, they need to find a different way to illustrate the changes.

[Figure]

**Response:** Good point. We have removed Figure 9.

**24)** Figure 10. It seems like simulations in Well A poorly represent the dynamic. Well B seems to have a systematic bias.

**Response:** At the first sight of figure 10, Well A poorly represents the dynamic and Well B seems to have a systematic bias. However, compared with the original MODFLOW-NWT set-up, which has a mean absolute error of 2.22 m between observed and simulated heads (Table 4), the errors between the observed and simulated head shown in figure 10 are much smaller. It is worth noting that the original MODFLOW-NWT set-up has been used in the management of water resources in the Hørring Municipality since 2009.

Modifications: We have rephrased the texts in lines 469-471 as follows:

Line 459-462: "There was generally a good agreement between the groundwater head level and dynamics simulated by SWAT-MODFLOW and that recorded at the two observation wells within the catchment, though the seasonal well drawdowns in Well A did not always occur in the observations (Fig. 7)."

**25)** The authors should include some model performance metrics when comparing the models.

**Response:** Actually, the comparison of model performance metrics of the models was included in Table 5 and also in the text between lines 438-446.

**References**

Bailey, R. T., Wible, T. C., Arabi, M., Records, R. M., and Ditty, J.: Assessing regional-scale spatio-temporal patterns of groundwater-surface water interactions using a coupled SWAT-MODFLOW model, Hydrological Processes, 30, 4420-4433, 10.1002/hyp.10933, 2016.

Gassman, P. W., Sadeghi, A. M., and Srinivasan, R.: Applications of the

SWAT Model Special Section: Overview and Insights, J Environ Qual, 43, 1-8, 10.2134/jeq2013.11.0466, 2014.

Guzman, J. A., Moriasi, D. N., Gowda, P. H., Steiner, J. L., Starks, P. J., Arnold, J. G., and Srinivasan, R.: A model integration framework for linking SWAT and MODFLOW, Environmental Modelling Software, 73, 103-116, 10.1016/j.envsoft.2015.08.011, 2015.

Lu, S., Andersen, H. E., Thodsen, H., Rubæk, G. H., and Trolle, D.: Extended SWAT model for dissolved reactive phosphorus transport in tile-drained fields and catchments, Agricultural Water Management, 175, 78-90, 10.1016/j.agwat.2015.12.008, 2016.

Neitsch, S. L., Arnold, J. G., Kiniry, J. R., and Williams, J. R.: Soil and water assessment tool theoretical documentation version 2009, Texas Water Resources Institute, 2011.

[Figure]

Norway

N

Sweden

Denmark

Germany

| | |
|---|---|
| 0 75 150 300 km | |

**Fig. 1.** Figure 1. Location of the Uggerby River catchment and Hjørring Municipality, and their delineation in SWAT and MODFLOW. The locations of wells distributed inside or outside the Uggerby River Catchment

- ● Drinking water wells
- ● Irrigation wells
- ● Wells outside
- ★ Hydrological stations and basin outlet

— SWAT streams
☐ Subbasins
High : 108.4 m a.s.l
Low : -0.2 m a.s.l
Active cells layer 1
Inactive cells layer 1

0  5  10  20 km

**Layer 1**

**Hydraulic conductivities (m s⁻¹)**

High : 0.0001

Low : 0

**Layer 2**

High : 0.000327

Low : 0

**Layer 3**

High : 0.0001

Low : 0

**Layer 4**

High : 1e-005

Low : 0

**Layer 5**

High : 0.0005

Low : 0

**Fig. 2.** Appendix A. The hydraulic conductivities of each cell in each layer in the steady-state MODFLOW-NWT set-up.

[Figure]

---

## Author Comment (AC2) · 17 Oct 2019

We thank the Anonymous Referee 2 for his/her thoughtful comments and efforts towards improving our manuscript.

**General comments:**

Liu et al. present a comparison of SWAT with SWAT-MODFLOW for the Uggerby catchment, Denmark. The study is well written with an excellent level of detail provided in the method. I have no working knowledge of the models applied in this study, so my

comments are high level, relating to the statistical interpretation of the results and their significance.

My main concern is that the conclusion that SWAT-MODFLOW is superior does not seem to be justified by model performances achieved in validation. Looking at Table 5, we see that the additional model flexibility offered by MODFLOW and MODFLOW with PEST significantly increase calibration performance without improving validation performance significantly. This suggests that these models are simply over-fitting. I don't think this paper requires any additional experiments to be run, but I do suggest that the results need to be interpreted accurately. Unless the authors offer a convincing reason for their current interpretation of the results (based on their validation), I would suggest that the whole discussion and conclusions need to be rewritten to be more reflective of an honest appraisal of the model performances. The other general issue is that the paper is very long. I think it can be shortened significantly without losing the key messages.

**Response:** We thank the reviewer for insightful comments. We have shortened the paper as much as we can (also in response to reviewer 1 comments). Regarding the model performance description, actually we evaluated the performance of the two models not only based on the statistic metrics values but also through visualization of hydrograph and according to the evaluation criteria recommended by Moriasi et al. (2015) which has widely been used for evaluating the performance of hydrological models. Although during the validation period, the percent bias (PBIAS) of SWAT-MODFLOW was slightly worse than SWAT; their values still fall in the same class according to the evaluation criteria recommended by Moriasi et al. (2015). However, according to the R2 and Nash–Sutcliffe efficiency (NSE) values, the evaluation criteria and visualization of the hydrograph (Fig. 6), we deem that the performance of SWAT-MODFLOW was overall better than SWAT. (Lines 445-453).

**Specific comments:**

[Figure]

**1)** Abstract - very long; considering shortening.

**Response:** Good suggestion. We have shortened it as much as we can, without compromising key outcomes.

**2)** Line 80 - no model considers the "entire" complexity. Please revise.

**Response:** Good point. We have rephrased the sentence in line 80 as follows:

Line 77-78: "In contrast, numerical, process-based models take into account more about the complexity and heterogeneity of river-aquifer systems."

**3)** Line 160 - please report

**Response:** Good suggestion. However, unfortunately, we have not found literature reporting the proportion of irrigation areas in Denmark. Instead, we found out that the annual irrigation amount during 1989-2007 was 175-259 million m$^3$ (Thorling et al., 2019).

Modifications: We have now added this information into the manuscript as follows:

Line 160-162: "In Denmark, approximately 800 million m$^3$ of water are abstracted annually and used for irrigation (175-259 million m$^3$ during 1989-2017) or drinking water(GEUS, 2009; Thorling et al., 2019), making the country highly dependent on groundwater."

**4)** Line 173 - The hypothesis that the "benefits of applying SWAT-MODFLOW outweigh the costs" is one that can be tested objectively (and is not answered in your results). I suggest reframing the study so that the aim is to explore the effects of introducing MODFLOW and MODFLOW pest into SWAT simulations in this particular catchment.

**Response:** The reviewer has raised a valid point. We have rephrased the hypothesis as follows: Line 28-31: We hypothesize that an integrated surface-subsurface model SWAT-MODFLOW performs better relative to a lumped semi-distributed catchment model SWAT when assessing the impacts of groundwater abstractions (for either

irrigation or drinking water) on streamflow patterns.

Line 174-176: We hypothesize that the SWAT-MODFLOW performs better relative to SWAT when assessing the impacts of groundwater abstractions (for either irrigation or drinking water) on streamflow patterns.

**5)** Line 325 - Is the water stress threshold taken as a single value for the whole catchment? If so, what are the limitations of this assumption? Would the threshold vary according to crop type/soil type?

**Response:** No. Actually three and four values for water stress threshold were taken for the whole catchment after calibration, as shown in Table 3 and Table 4.

**6)** Page 12 - the reader does not need to know the names of your python scripts.

**Response:** Good point. We have now deleted the names of our python scripts.

**7)** Figure 5 - not particularly helpful. I think this can be omitted.

**Response:** Good suggestion. We have now deleted figure 5.

**8)** Line 478 (and throughout the results section)... lots of results reported in vague terms ("little higher", "much lower" ... etc). Please report

**Response:** Good suggestions! As suggested, we have replaced the vague terms by reporting

**9)** Line 552 - it does not reflect a shortcoming of the SWAT groundwater module if the improvements are simply overfitting.

**Response:** We have now made it clear that we refer to a shortcoming in the conceptual model of SWAT, as it ignores the variability in distributed parameters such as hydraulic conductivity and storage coefficients, lumps spatial detail within the groundwater domain of a subbasin, and contributes to the stream network as baseflow based on a linear reservoir approximation.

Modifications: Line 545-548: "This reflects the shortcoming of the concept for SWAT groundwater module, which ignores the variability in distributed parameters such as hydraulic conductivity and storage coefficients, represents groundwater by a lumped module in individual subbasins, and contributes to the stream network as baseflow based on a linear reservoir approximation."

**10)** Line 560 - this conclusion is not warranted if the model has been overfitted, as is suggested by results reported in Table 5.

**Response:** We do not think the model was overfitted, but we made some changes in Line 560 to make the conclusion more convincing as follows:

Line 551-553: "The availability of spatial-temporal patterns of the groundwater head in SWAT-MODFLOW could significantly benefit groundwater resources management and provide the spatial explicitly water resources dynamics within a catchment."

**11)** Figure 6 - remove background shading.

**Response:** The reviewer has given us a good suggestion, but unfortunately, we are not able to remove the background shading in figure 6. There are some zone boundary lines inside the original shape file of layers (layer 1 and layer 3) showing the zones of different hydrological properties. If we remove the shading, those zone boundary lines will show up and then mix up with the contours, making the figure more blurry. Instead, the readers can zoom in the figure to read the figure more clearly.

**12)** Figure 10 - These are not promising results. Seasonal well drawdowns in the simulations do not occur in the observations. Why should this not be reported as evidence of the poor performance of SWAT-MODFLOW?

**Response:** Good point. We have rephrased the texts in lines 469-471 as follows:

Lines 469-471: There was generally a good agreement between the groundwater head level and dynamics simulated by SWAT-MODFLOW and that recorded at the two observation wells within the catchment, though the seasonal well drawdowns in Well A

did not always occur in the observations (Fig. 10).

**13)** Lastly, the arbitrary labels attached to NSE scores ("satisfactory" etc) are inappropriate. Report the numbers, show the data, and let the reader decide what is satisfactory.

**Response:** Actually, the labels attached to the statistical metrics are not arbitrary. We evaluated them according to the criteria recommended by (Moriasi et al., 2015), which has been widely used to evaluate the performance of hydrological models.

**References**

Moriasi, D. N., Gitau, M. W., Pai, N., and Daggupati, P.: Hydrologic and water quality models: Performance measures and evaluation criteria, Transactions of the ASABE, 58, 1763-1785, 2015.

Thorling, L., Albers, C., Ditlefsen, C., Ernstsen, V., Hansen, B., Johnsen, A., and Troldborg, L.: Grundvandsovervågning, Status og udvikling 1989–2017, GEUS, De Nationale Geologiske Undersøgelser for Danmark og Grønland Energi-, Forsynings- og Klimaministeriet, Copenhagen, 140, 2019.

---

## Author Comment (AC3) · 22 Oct 2019

**Appendix A.** The hydraulic conductivities (m s$^{-1}$) of sedimentary materials in each layer in the steady-state MODFLOW-NWT set-up.